# AI-based discovery and cryoEM structural elucidation of a K$_{ATP}$ channel pharmacochaperone

**Assmaa Elsheikh[1,2], Camden M Driggers[1], Ha H Truong[3], Zhongying Yang[1], John Allen[1], Niel M Henriksen[3], Katarzyna Walczewska-Szewc[4], Show-Ling Shyng[1]\***

[1]Department of Chemical Physiology and Biochemistry, Oregon Health & Science University, Portland, United States; [2]Department of Medical Biochemistry, College of Medicine, Tanta University, Tanta, Egypt; [3]Atomwise Inc, San Francisco, United States; [4]Institute of Physics, Faculty of Physics, Astronomy and Informatics, Nicolaus Copernicus University in Toruń, Toruń, Poland

**\*For correspondence:**
shyngs@ohsu.edu

## eLife Assessment

This **important** study demonstrates that screening by artificial intelligence can identify relevant novel compounds for interacting with KATP channels. The experimental work is **compelling**. The broader significance of this work relates to the possibility that KATP channel mutations linked to congenital hyperinsulinism may be effectively rescued to the cell surface with a drug, which could normalize insulin secretion or enhance the effectiveness of existing KATP channel activators such as diazoxide.

**Abstract** Pancreatic K$_{ATP}$ channel trafficking defects underlie congenital hyperinsulinism (CHI) cases unresponsive to the K$_{ATP}$ channel opener diazoxide, the mainstay medical therapy for CHI. Current clinically used K$_{ATP}$ channel inhibitors have been shown to act as pharmacochaperones and restore surface expression of trafficking mutants; however, their therapeutic utility for K$_{ATP}$ trafficking-impaired CHI is hindered by high affinity binding, which limits functional recovery of rescued channels. Recent structural studies of K$_{ATP}$ channels employing cryo-electron microscopy (cryoEM) have revealed a promiscuous pocket where several known K$_{ATP}$ pharmacochaperones bind. The structural knowledge provides a framework for discovering K$_{ATP}$ channel pharmacochaperones with desired reversible inhibitory effects to permit functional recovery of rescued channels. Using an AI-based virtual screening technology AtomNet followed by functional validation, we identified a novel compound, termed Aekatperone, which exhibits chaperoning effects on K$_{ATP}$ channel trafficking mutations. Aekatperone reversibly inhibits K$_{ATP}$ channel activity with a half-maximal inhibitory concentration (IC$_{50}$) ~9 µM. Mutant channels rescued to the cell surface by Aekatperone showed functional recovery upon washout of the compound. CryoEM structure of K$_{ATP}$ bound to Aekatperone revealed distinct binding features compared to known high affinity inhibitor pharmacochaperones. Our findings unveil a K$_{ATP}$ pharmacochaperone enabling functional recovery of rescued channels as a promising therapeutic for CHI caused by K$_{ATP}$ trafficking defects.

## Introduction

Pancreatic β-cells secrete insulin in response to an increase in blood glucose levels to maintain glucose homeostasis. Central to this process is the ATP-sensitive potassium (K$_{ATP}$) channel assembled from four pore-forming Kir6.2 subunits (encoded by *KCNJ11*) and four regulatory SUR1 subunits (encoded

by *ABCC8*) (*Aguilar-Bryan and Bryan, 1999*; *Ashcroft, 2023*; *Driggers and Shyng, 2023*; *Nichols, 2006*). $K_{ATP}$ channels are gated by intracellular ATP and ADP, the concentrations of which change following glucose metabolism. This enables $K_{ATP}$ channels to couple blood glucose levels with β-cell membrane potential to control insulin secretion (*Ashcroft, 2023*; *Merrins et al., 2022*; *Nichols et al., 2022*; *Thompson and Satin, 2021*). When serum blood glucose levels rise, the intracellular ATP/ADP ratio increases, which promotes $K_{ATP}$ channel closure, membrane depolarization, $Ca^{2+}$ influx through voltage-gated $Ca^{2+}$ channels, and insulin secretion. Conversely, a fall in serum glucose reduces the intracellular ATP/ADP ratio, which favors opening of $K_{ATP}$ channels, leading to membrane hyperpolarization and termination of insulin secretion. Underscoring the importance of β-cell $K_{ATP}$ channels in glucose regulation, gain-of-function $K_{ATP}$ mutations cause neonatal diabetes, while loss-of-function $K_{ATP}$ mutations cause congenital hyperinsulinism (CHI) (*Ashcroft, 2023*; *De Franco et al., 2020*; *ElSheikh and Shyng, 2023*; *Nichols et al., 2022*; *Pipatpolkai et al., 2020*). $K_{ATP}$ inhibitors such as sulfonylureas and glinides or activators such as diazoxide may be used to treat diseases caused by gain- or loss-of-function $K_{ATP}$ mutations, respectively; however, effectiveness of these existing drugs is highly dependent on the underlying mutant channel defects (*ElSheikh and Shyng, 2023*; *Martin et al., 2020*; *Pipatpolkai et al., 2020*; *Rosenfeld et al., 2019*). A case in point is CHI caused by mutations in *ABCC8* or *KCNJ11* that prevent proper folding and assembly of the channel proteins in the endoplasmic reticulum (ER) and subsequent trafficking of $K_{ATP}$ channels to the β-cell plasma membrane (*Cartier et al., 2001*; *Crane and Aguilar-Bryan, 2004*; *ElSheikh and Shyng, 2023*; *Fukuda et al., 2011*; *Yan et al., 2004*; *Yan et al., 2007*). Individuals harboring such mutations present severe CHI disease unresponsive to diazoxide and often require pancreatectomy to prevent life-threatening hypoglycemia. For these patients, drugs that overcome $K_{ATP}$ channel trafficking defects are needed (*ElSheikh and Shyng, 2023*).

Pharmacological chaperones (pharmacochaperones) are small molecules that bind to target proteins and facilitate their folding, assembly, and trafficking (*Bernier et al., 2004*; *Convertino et al., 2016*; *Leidenheimer and Ryder, 2014*). Previous work has shown that $K_{ATP}$ channel inhibitors act as pharmacochaperones to overcome $K_{ATP}$ trafficking defects (*Martin et al., 2020*). Among these inhibitor pharmacochaperones, glibenclamide (GBC), a high affinity sulfonylurea, and repaglinide (RPG), a high affinity glinide, have long been used clinically for type 2 diabetes (*Sahin et al., 2024*); in addition, carbamazepine (CBZ), an anticonvulsant known to inhibit voltage-gated $Na^+$ channel, has been found to potently inhibit $K_{ATP}$ channels (*Chen et al., 2013*; *Devaraneni et al., 2015*; *Martin et al., 2016*; *Yan et al., 2004*; *Yan et al., 2006*). Kir6.2 is a transmembrane (TM) protein with two TM helices and cytoplasmic N- and C-termini. SUR1, a member of the ABC protein family, consists of a canonical ABC core structure of two 6-TM helix TM domains (TMD), TMD1 and TMD2, and two cytoplasmic nucleotide binding domains (NBDs), NBD1 and NBD2; in addition, it contains an N-terminal TM domain TMD0 which is connected to the ABC core via a cytoplasmic linker, L0 (*Driggers and Shyng, 2023*). Recent high-resolution cryo-electron microscopy (cryoEM) structures of $K_{ATP}$ channels along with functional studies revealed that $K_{ATP}$ inhibitors bind at a common pocket in the TMD of the SUR1-ABC core that serves to anchor and immobilize the Kir6.2 N-terminus, which interaction stabilizes nascent channel assembly and hence trafficking to the cell surface (*Martin et al., 2017a*; *Martin et al., 2020*; *Martin et al., 2019*; *Martin et al., 2017b*). Paradoxically, the same SUR1-Kir6.2 interaction also results in channel closure, explaining the dual pharmacochaperone and inhibitory effects of the aforementioned compounds. Thus, a current challenge in harnessing the pharmacochaperone effects of the above known $K_{ATP}$ inhibitors to treat CHI caused by $K_{ATP}$ trafficking mutations is their relatively high binding affinity, which impedes channel functional recovery post-rescue (*ElSheikh and Shyng, 2023*). Inhibitor pharmacochaperones with lower affinity for $K_{ATP}$ channels would allow their rapid dissociation from rescued channels at the cell surface upon pharmacochaperone washout for channel function.

$K_{ATP}$ channel drug discovery to date has historically involved serendipitous findings, modification of existing scaffolds with medicinal chemistry, or costly functional screening of large compound libraries without knowledge of the drug binding site and mechanism (*Kharade et al., 2016*; *Nichols, 2023*). The structural insights on existing $K_{ATP}$ pharmacochaperones described above provide a framework for structure-based discovery of new $K_{ATP}$ channel pharmacochaperones. Recently, we conducted a virtual screening study against an RPG-bound $K_{ATP}$ channel structure using AtomNet (*Atomwise AIMS Program, 2024*), a deep neural network for structure-based drug design and discovery platform, developed by Atomwise Inc, to search for new compounds that bind to the $K_{ATP}$ pharmacochaperone

binding pocket (**Atomwise AIMS Program, 2024**; **Stafford et al., 2022**; **Wallach et al., 2015**). Specifically, we focused on compounds that can rescue $K_{ATP}$ trafficking mutants without irreversibly inhibiting their function. Here, we report the identification and characterization of a novel $K_{ATP}$ modulator that exhibits pharmacochaperone and reversible inhibitory effects on SUR1/Kir6.2 channels, which we termed Aekatperone (AE-, the initials of the first author of this study, -katperone, $K_{ATP}$ pharmacochaperone). CryoEM structure of SUR1/Kir6.2 $K_{ATP}$ channel in complex with Aekatperone reveals the compound in the predicted binding pocket but with distinct binding features compared to the previously published high affinity inhibitor pharmacochaperones. Our study provides proof of principle for structure-based $K_{ATP}$ channel drug discovery to expand $K_{ATP}$ pharmacology toward mechanism-based and individualized medicine.

## Results

### Virtual screening for new $K_{ATP}$ channel binders using the AtomNet technology

CryoEM structures of $K_{ATP}$ channels in complex with pharmacochaperones afford drug discovery opportunities based on structural insights (**Ding et al., 2019**; **Martin et al., 2020**; **Martin et al., 2019**; **Sung et al., 2022**). Using an Artificial Intelligence (AI) virtual screening system AtomNet developed by Atomwise (**Atomwise AIMS Program, 2024**; **Gniewek et al., 2021**; **Wallach et al., 2015**), we performed virtual screening of ~2.5 million small organic molecules in the Enamine in-stock library against the binding site of a high affinity $K_{ATP}$ inhibitor pharmacochaperone, RPG, resolved in our published RPG-bound $K_{ATP}$ channel (hamster SUR1 plus rat Kir6.2) cryoEM structure (PDB ID 6PZ9; ~3.65 Å) (**Atomwise AIMS Program, 2024**; **Martin et al., 2019**).

The RPG binding site being screened involves the following residues: R306, Y377, A380, I381, W430, F433, L434, N437, M441, T588, L592, L593, S595, V596, T1242, N1245, R1246, E1249, W1297, R1300 of the SUR1 subunit (chain C in PDB ID 6PZ9) and residues M1, L2, S3, K5, G6, I7 of Kir6.2 (chain D in PDB ID 6PZ9) (**Figure 1A**). RPG and all water molecules were removed from the site before screening. Each compound was scored by AtomNet and the compounds were ranked by their scores (**Atomwise AIMS Program, 2024**; **Stafford et al., 2022**). Following diversity clustering to minimize selection of structurally similar scaffolds and filtering for oral availability as well as excluding toxicophores (see Methods for details), the top 96 predicted binders (**Supplementary file 1**) were subjected to subsequent biochemical and functional testing as described below.

### Testing top-ranking compounds from virtual screening for pharmacochaperone and gating effects on pancreatic $K_{ATP}$ channels

The study aims to identify new $K_{ATP}$ channel pharmacochaperones that will lead to surface expression and functional recovery of CHI-causing, trafficking-impaired $K_{ATP}$ channels. Accordingly, our initial screening employed a western blot assay that tests the ability of compounds to restore the maturation of SUR1 trafficking mutants. SUR1 contains two N-linked glycosylation sites which are core-glycosylated in the ER. Upon co-assembly with Kir6.2 into a full $K_{ATP}$ complex and its subsequent trafficking through the trans Golgi network, SUR1 becomes complex glycosylated. In western blots, the core- and complex-glycosylated SUR1 proteins appear as a lower and an upper band (**Raab-Graham et al., 1999**; **Zerangue et al., 1999**), which are referred to as immature and mature SUR1, respectively. Mutant channel proteins that fail to fold or assemble correctly are retained in the ER such that SUR1 is unable to mature, resulting in an absence of the SUR1 upper band in western blot (**Martin et al., 2016**; **Yan et al., 2004**; **Yan et al., 2006**; **Yan et al., 2007**). A great majority of such trafficking mutations is in the N-terminal TMD0 domain of SUR1, which is the primary assembly domain with Kir6.2 (**Martin et al., 2020**). Most of these mutations are responsive to pharmacochaperone rescue. A well-characterized example is F27S-SUR1 ($SUR1_{F27S}$) (**Yan et al., 2007**), which hinders SUR1 from acquiring the mature upper band; however, overnight treatment of cells with previously identified pharmacochaperones such as GBC restores the upper band (**Yan et al., 2007**). We therefore chose $SUR1_{F27S}$ for our initial pharmacochaperone screening (**Figure 1—figure supplement 1A**).

To minimize protein expression variations and increase screening efficiency, we transduced rat insulinoma cells INS-1 in a large culture dish with recombinant adenoviruses encoding $SUR1_{F27S}$ (from golden hamster *Mesocricetus auratus*) and wild-type (WT) Kir6.2 (from rat *Rattus rattus*), and then

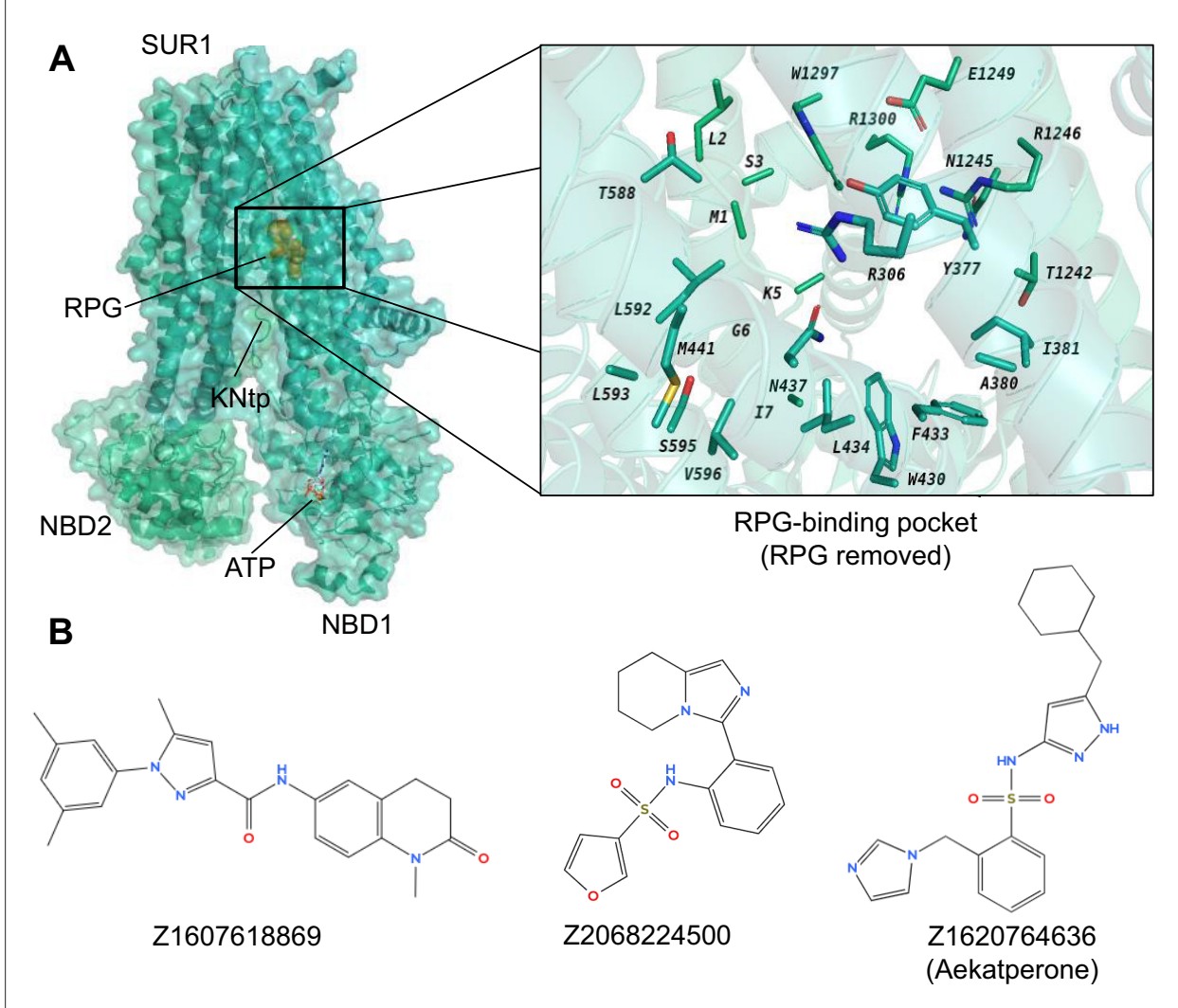

**Figure 1.** Identification of potential pharmacochaperones for pancreatic K_ATP channels through virtual screening and experimental validation. (**A**) *Left*: Cryo-electron microscopy (cryoEM) structure of repaglinide (RPG)-bound pancreatic K_ATP channel (PDB ID: 6PZ9; only the SUR1 ABC core and Kir6.2-N terminal peptide [KNtp] are shown; RPG is shown in gold spheres and ATP bound at the nucleotide binding domain [NBD]1 of SUR1 is shown as red sticks). *Right*: A close-up view of the target site used for virtual screening of a library of ~2.5 million compounds (from Enamine), utilizing the AtomNet model (RPG is removed to show only the binding residues from SUR1). From this screening, a set of 96 'top scoring compounds' was selected for experimental validation. (**B**) Chemical structures of the top three compounds, C24, C27, and C45, identified via experimental testing of pharmacochaperone effects on a K_ATP trafficking mutant (*Figure 1—figure supplement 1*) correspond to ZINC IDs Z1607618869, Z2068224500, and Z1620764636, respectively. These compounds are chemically distinct from one another. Note C27 and C45 both contain a sulfonamide group (-SO2-N-), which is different from the sulfonylurea group (-SO2-NH-CO-NH-) seen in known K_ATP inhibitors such as glibenclamide and tolbutamide.

The online version of this article includes the following source data and figure supplement(s) for figure 1:

**Figure supplement 1.** Experimental testing of the top 96 compounds identified from AtomNet-based virtual screening.

**Figure supplement 1—source data 1.** PDF file containing original western blots for panel B, indicating the relevant bands and treatments.

**Figure supplement 1—source data 2.** Original files for western blot analysis displayed in panel B.

**Figure supplement 2.** The top three compounds identified via western blot screening assay inhibit K_ATP channel activity.

divided the transduced cells into multi-well tissue culture plates for compound testing (see Methods and *Figure 1—figure supplement 1A*). Cells were treated overnight (16 hr) with either DMSO (vehicle control), 10 µM of GBC (positive control), or 10 µM of each of the 96 test compounds. SUR1_{F27S} was analyzed by western blot. As expected, in cells treated with DMSO only the immature lower SUR1_{F27S} band was detected; GBC treatment led to the appearance of a strong mature SUR1_{F27S} band (*Figure 1—figure supplement 1B*). (Note the low levels of endogenous SUR1 in INS-1 cells was not

detectable at the exposure used to detect exogenous $SUR1_{F27S}$.) Treatment with some of the test compounds yielded weak or moderate upper $SUR1_{F27S}$ band, suggesting pharmacochaperone effects. We selected three compounds that showed consistent results in two independent screening experiments (*Figure 1—figure supplement 1B*) for further testing in a different assay.

The three compounds selected, C24, C27, and C45, correspond to ZINC database (https://zinc.docking.org/) ID: Z1607618869, Z2068224500, and Z1620764636, respectively, and represent diverse chemical scaffolds, as shown in *Figure 1B*. Since all known $K_{ATP}$ pharmacochaperones to date also inhibit channel activity, we tested whether the three compounds likewise act as $K_{ATP}$ inhibitors using electrophysiology. For this and subsequent experiments that do not require screening of a large number of compounds, COSm6 cells lacking endogenous $K_{ATP}$ channels were used for transient transfection of WT or mutant $K_{ATP}$ channels. Inside-out patch-clamp recordings of cells expressing $K_{ATP}$ channels (hamster WT SUR1+rat WT Kir6.2) showed that indeed, all three compounds inhibited $K_{ATP}$ channel currents (*Figure 1—figure supplement 2A–C*). Upon exposure to the compounds, an immediate reduction in current amplitudes were observed, which was followed by a steady further decline in channel activity. To test reversibility of the inhibitory effects, membrane patches were returned to the K-INT solution without compounds and current recovery was monitored (*Figure 1—figure supplement 2Ai–Ci*). Data from these initial studies allowed for a rough estimate of the potency and reversibility of these compounds. Of the three compounds Z1620764636 (C45) showed a clear dose-dependent inhibition, with an estimated steady-state $IC_{50}$ between 1 and 10 µM, and clear current recovery upon compound washout (*Figure 1—figure supplement 2Ci*). Compound Z1607618869 (C24) at 5 µM showed >50% steady-state inhibition and reversibility but increasing the concentrations did not yield consistent further inhibition (*Figure 1—figure supplement 2Aii*), which could be due to its poor solubility apparent from visual inspection. Compound Z2068224500 (C27) is the least potent, with an estimated steady-state $IC_{50} \sim 20$–50 µM, and also the least reversible (*Figure 1—figure supplement 2Bi*). Based on these results, we chose to focus further studies on C45, which we named Aekatperone, and will be referred to as such hereinafter.

## Aekatperone: a novel $K_{ATP}$ modulator with dual pharmacochaperone and inhibitory actions on pancreatic $K_{ATP}$ channels

Our results from the initial screening experiments suggest Aekatperone as a potential reversible inhibitor pharmacochaperone for surface expression rescue of trafficking-impaired $K_{ATP}$ mutants. The observed current recovery following Aekatperone washout predicts functional recovery of the mutant channels post-treatment. Further biochemical, functional, and structural studies were conducted with Aekatperone obtained from Enamine.

First, we tested the pharmacochaperone effects of Aekatperone on several other trafficking mutations: $SUR1_{A30T}$, $SUR1_{A116P}$, and $SUR1_{V187D}$. These mutations are among the ones that have been shown to not affect the functional properties of $K_{ATP}$ channels (*Martin et al., 2016*; *Yan et al., 2004*). COSm6 cells transfected with mutant SUR1 and WT Kir6.2 were treated with DMSO vehicle control or Aekatperone for 16 hr, and mutant SUR1 maturation was monitored by western blotting. Consistent with published studies, western blot of mutants from DMSO-treated cells showed clear lower immature band but no or barely detectable upper mature band (*Figure 2A*). Overnight treatment with 100 µM Aekatperone led to a clear increase of the mature upper band (*Figure 2A*). A detailed dose-response for $SUR1_{A30T}$ showed that the pharmacochaperone effect of Aekatperone was already evident at 10 µM and became more pronounced as the concentration increased, with the effect plateauing between 100 and 200 µM (*Figure 2B*). At 200 µM, the intensity of the upper band relative to the total intensity of both lower and upper bands for $SUR1_{A30T}$ was 64.21 ± 3.35% of that observed for WT SUR1 (n=3, p=0.0133 by Student's t-test) based on densitometry by ImageJ (*Figure 2B*). These results support Aekatperone as a general SUR1-TMD0 $K_{ATP}$ pharmacochaperones.

To demonstrate that the increased mature mutant SUR1 band upon Aekatperone treatment reflects increased surface expression of the mutant channels, we directly monitored $K_{ATP}$ channel surface expression by immunofluorescent staining. In these experiments, COSm6 cells were co-transfected with Kir6.2 and N-terminally FLAG-tagged SUR1. Surface $K_{ATP}$ channels can be detected by incubating cells in medium containing anti-FLAG antibody without cell permeation. As expected, cells transfected with the WT $K_{ATP}$ channel cDNAs showed robust surface staining. In contrast, cells transfected with $SUR1_{F27S}$ or $SUR1_{A30T}$ cDNAs and treated overnight with DMSO showed little surface staining.

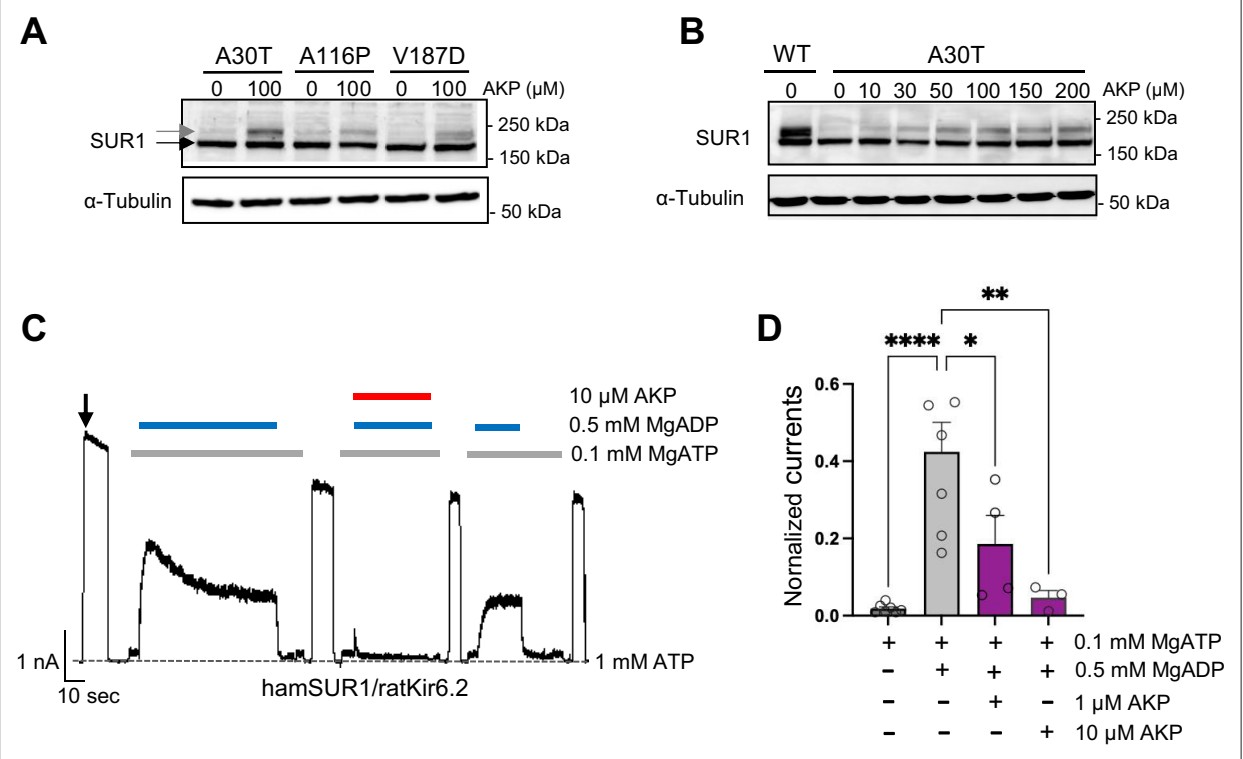

**Figure 2.** Aekatperone has dual pharmacochaperone and inhibitory actions on pancreatic $K_{ATP}$ channels. (**A**) Representative western blots of SUR1 from COSm6 cells co-transfected with cDNAs of wild-type (WT) Kir6.2 and trafficking mutants of SUR1 TMD0 domain, A30T, A116P, or V187D, and treated with either 0.1% DMSO (0 μM) or 100 μM Aekatperone (AKP) for 16 hr. The core-glycosylated immature SUR1 and the complex-glycosylated mature SUR1 are indicated by the black and gray arrows, respectively. The tubulin blot below serves as a loading control. (**B**) Representative western blots of SUR1 from COSm6 cells co-transfected with cDNAs of WT Kir6.2 and a SUR1 trafficking mutant A30T, and treated with either 0.1% DMSO (0 μM) or various concentrations of Aekatperone (10, 50, 100, 150, 200 μM) for 16 hr. WT SUR1 from cells co-transfected with WT Kir6.2 and WT SUR1 without Aekatperone treatment served as a control (left lane). (**C**) Representative recording from COSm6 cells co-transfected with hamster SUR1 and rat Kir6.2. Channels were exposed to K-INT solution upon patch excision (arrow) and exposed to solutions containing MgATP, MgADP, or Aekatperone as indicated by the bars above the recordings and the labels on the right. The patch was exposed to 1 mM ATP periodically to ensure the baseline has not shifted (gray dashed line). (**D**) Quantification of currents (normalized to currents in K-INT/1 mM EDTA at the time of patch excision) in various solutions from recordings such as that shown in (**A**). Each bar represents the mean ± SEM of at least three patches, with circles showing individual patches. *p<0.05 by one-way ANOVA and Dunnet's post hoc test.

The online version of this article includes the following source data and figure supplement(s) for figure 2:

**Source data 1.** PDF file containing original western blots for panels A and B, indicating the relevant bands and treatments.

**Source data 2.** Original files for western blot analysis displayed in panels A and B.

**Figure supplement 1.** Aekatperone rescues surface expression of SUR1F27S and SUR1A30T trafficking mutants.

**Figure supplement 2.** Aekatperone has the same pharmacochaperoning and inhibitory effects on human $K_{ATP}$ channels.

**Figure supplement 2—source data 1.** PDF file containing original western blots for panel A, indicating the relevant bands and treatments.

**Figure supplement 2—source data 2.** Original files for western blot analysis displayed in panel A.

However, staining of fixed and membrane permeabilized cells transfected with the SUR1$_{F27S}$ or SUR1$_{A30T}$ mutant revealed abundant intracellular fluorescence signal concentrated in the perinuclear region consistent with the mutations causing ER retention of channel proteins (*Figure 2—figure supplement 1*, lower panels). Upon Aekatperone treatment, there was a clear increase in the percentage of cells showing surface staining of SUR1$_{F27S}$ or SUR1$_{A30T}$ mutant channels (*Figure 2—figure supplement 1*, top panels), providing direct evidence that Aekatperone rescues surface expression of CHI-causing SUR1 trafficking mutations.

In the initial screening, we found that Aekatperone inhibited $K_{ATP}$ currents in inside-out patch-clamp recording experiments (*Figure 1—figure supplement 2C*). We further investigated the effects of Aekatperone on $K_{ATP}$ channel response to MgADP stimulation, which is required for physiological

activation of $K_{ATP}$ channels in response to glucose deprivation-induced increases in intracellular ADP/ATP ratios (**Nichols et al., 1996**; **Shyng et al., 1998**). Increased MgADP concentrations promote NBD dimerization of SUR1 and increase channel opening (**Driggers and Shyng, 2023**; **Wang et al., 2022**; **Wu et al., 2018**; **Zhao and MacKinnon, 2021**). Because known $K_{ATP}$ inhibitors such as GBC and RPG prevent NBD dimerization by their binding inside a common SUR1 pocket (**Ding et al., 2019**; **Lee et al., 2017**; **Martin et al., 2019**; **Martin et al., 2017b**; **Wu et al., 2018**), against which the virtual screen leading to Aekatperone was conducted, we anticipated that Aekatperone may likewise diminish $K_{ATP}$ stimulation by MgADP. As shown in **Figure 2C and D**, robust activation of $K_{ATP}$ channel activity was observed upon addition of 0.5 mM MgADP to the 0.1 mM inhibitory ATP concentration (see Methods) as reported previously (**Yan et al., 2004**). Inclusion of 10 µM Aekatperone occluded the stimulatory effect of MgADP, but MgADP stimulation recovered upon subsequent removal of Aekatperone (**Figure 2C**). Thus, like GBC and RPG, Aekatperone blocks channel response to MgADP (**Dabrowski et al., 2001**; **Yan et al., 2006**); however, unlike GBC and RPG, Aekatperone's effect is readily reversible. Importantly, the pharmacochaperone effects of Aekatperone on trafficking mutants and reversible inhibitory effect of Aekatperone on channel response to MgADP were also observed in channels formed by human SUR1 and human Kir6.2 (**Figure 2—figure supplement 2**), which are highly homologous to hamster SUR1 and rat Kir6.2 at the protein level (95% and 96%, respectively). These results demonstrate that Aekatperone is also effective on human $K_{ATP}$ channels.

## SUR1 trafficking mutants rescued by Aekatperone show functional recovery upon removal of the compound

The ability of $K_{ATP}$ channels to open in response to hypoglycemia is critical to preventing inappropriate insulin secretion. Based on in vitro electrophysiological experiments showing the reversible inhibitory effect of Aekatperone (**Figure 1—figure supplement 2C**, **Figure 2C and D**), we hypothesize that trafficking mutants rescued to the cell surface by the drug will open in response to metabolic inhibition, a condition that mimics hypoglycemia, once the drug is removed.

To test the above hypothesis, we employed an $Rb^+$ efflux assay that allows assessment of channel activity in intact cells (**ElSheikh et al., 2024**). In this assay, $Rb^+$, which passes through $K_{ATP}$ channels, is used as a reporter ion to monitor $K_{ATP}$ activity. COSm6 cells transiently expressing WT $K_{ATP}$ channels were preloaded with $Rb^+$ and treated with metabolic inhibitors (1 mM deoxy-glucose plus 2.5 µg/ml oligomycin, see Methods), which activate $K_{ATP}$ channels by lowering intracellular ATP/ADP ratios and mimic hypoglycemia. $Rb^+$ efflux during a 30 min incubation period was measured in the absence or presence of a range of Aekatperone concentrations (1–200 µM). As shown in **Figure 3A**, Aekatperone dose-dependently reduced $Rb^+$ efflux, with an $IC_{50}$ of 9.23 µM±0.36 µM (**Figure 3B**). These results are consistent with the electrophysiology data showing inhibition of $K_{ATP}$ currents and MgADP response by Aekatperone in isolated membrane patches (**Figure 1—figure supplement 2C**, **Figure 2C**). Of note, to ensure Aekatperone is the active compound in our biochemical and functional assays, we tested Aekatperone synthesized from a different commercial source (Acme Bioscience, Palo Alto, CA, USA) and confirmed its effects on $K_{ATP}$ channels (**Figure 3—figure supplement 1**).

Having established the acute inhibitory effect of Aekatperone on the response of WT $K_{ATP}$ channels to metabolic inhibition, we next asked whether trafficking mutant channels rescued by overnight Aekatperone treatment would show functional recovery following Aekatperone removal. To test this, COSm6 cells transfected with $SUR1_{F27S}$ and $SUR1_{A30T}$ mutant plasmids were treated for 16 hr with Aekatperone at 10, 30, 50, 100, or 200 µM, or 0.1% DMSO as a vehicle control. Cells were then washed in medium lacking Aekatperone for 30 min before being subjected to $Rb^+$ efflux assays in the presence of metabolic inhibitors (see Methods). Under these conditions, we observed for both mutants a significant increase in $Rb^+$ efflux in cells treated with Aekatperone in a dose-dependent manner compared with cells treated with 0.1% DMSO (**Figure 3C and D**). This is in stark contrast to cells treated overnight with the high affinity inhibitor pharmacochaperone GBC followed by 30 min incubation in medium lacking GBC, where $Rb^+$ efflux was below that seen in cells treated overnight with the DMSO vehicle control (note in DMSO-treated controls, there were still low levels of surface expression of the mutant channels to give rise to the low levels of efflux).

Diazoxide is a $K_{ATP}$ opener and the only pharmacological treatment for CHI. We further evaluated the diazoxide response of mutant $K_{ATP}$ channels rescued by Aekatperone using $Rb^+$ efflux assay. Because previous studies have shown that diazoxide has no chaperoning activity and it actually

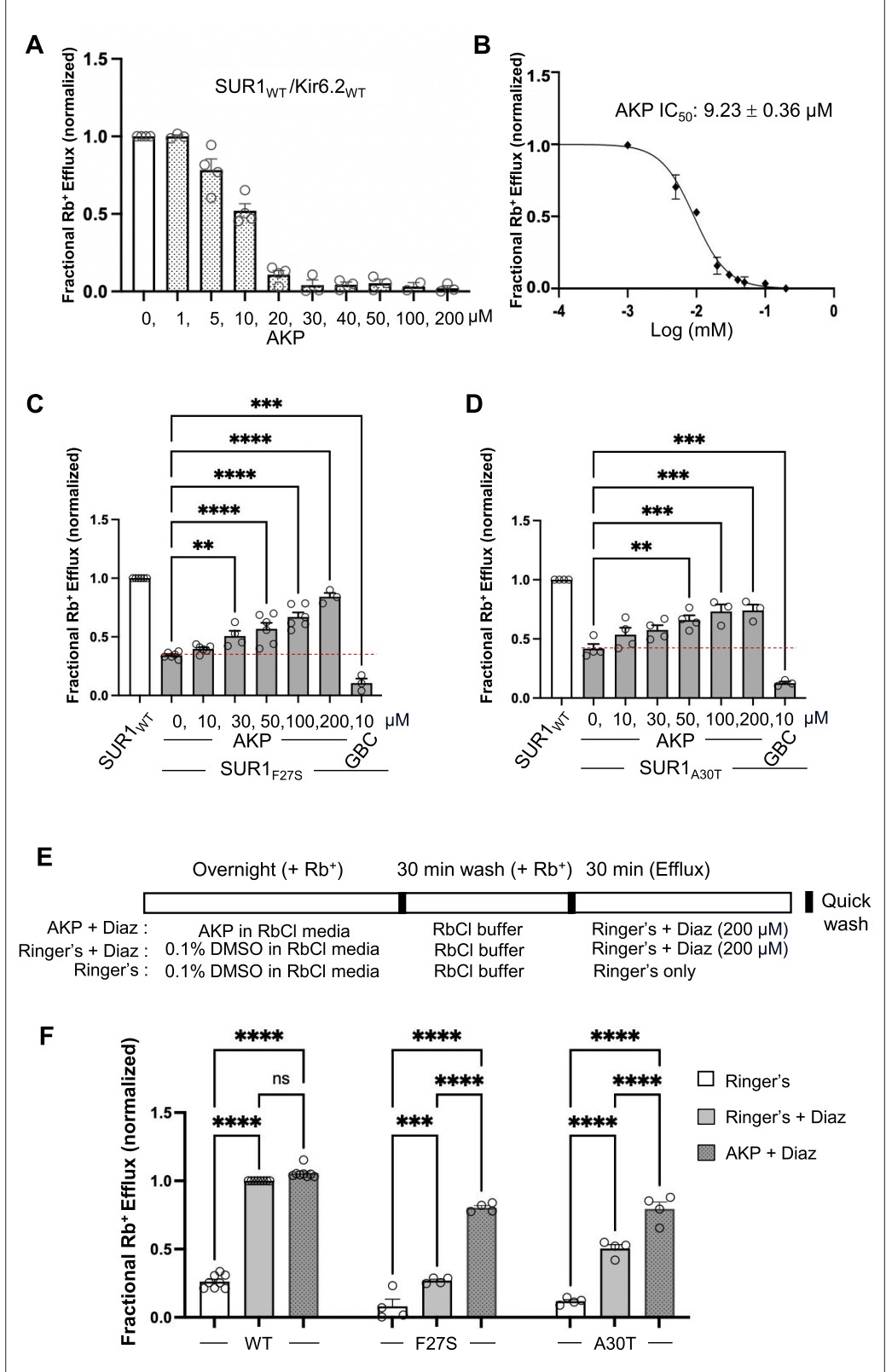

**Figure 3.** Functional recovery of mutant $K_{ATP}$ channels rescued by Aekatperone. (**A**) $Rb^+$ efflux assay results showing Aekatperone (AKP) dose-dependently inhibited wild-type (WT) pancreatic $K_{ATP}$ channels (hamster SUR1 and rat Kir6.2) expressed in COSm6 cells and opened by metabolic inhibition (see Methods). The fractional $Rb^+$ efflux was calculated by subtracting efflux in untransfected cells and normalizing to efflux in cells treated with 0.1% DMSO.

*Figure 3 continued on next page*

*Figure 3 continued*

(**B**) Dose-response curve of Aekatperone inhibition from data shown in (**A**) fitted with a Hill equation with variable slope using GraphPad Prism 10. The IC$_{50}$ is 9.23 μM±0.36 μM. Error bars represent the SEM. (**C, D**) Bar graphs showing dose-response enhancement in K$_{ATP}$ channel activity as assessed by Rb$^+$ efflux assay in COSm6 cells expressing two different trafficking mutations, SUR1$_{F27S}$ (**A**) and SUR1$_{A30T}$ (**B**). The cells were treated with varying concentrations of Aekatperone (10, 30, 50, 100, or 200 μM), glibenclamide (GBC) at 10 μM, or 0.1% DMSO as a vehicle control (0 μM Aekatperone) for 16 hr. Aekatperone and GBC was excluded from the efflux solutions during the efflux assay. Untransfected (UT) cells were included to quantify background Rb$^+$ efflux, which was subtracted from other experimental readings. The data were normalized to the fractional Rb$^+$ efflux of cells expressing WT channels. Error bars represent the SEM of at least three independent experiments (circles are individual data points from three to six different experiments). Statistical significance was performed using one-way ANOVA followed by Dunnett's post hoc multiple comparison test, alpha = 0.05. *p<0.05, **p<0.01, ***p<0.001, ****p<0.0001. A red dashed line is shown to indicate the basal efflux of the mutant channels under vehicle control conditions (in the absence of Aekatperone). (**E**) Schematic of experimental design for (**F**). COSm6 cells transfected with WT or various mutant channels were treated with 0.1% DMSO (vehicle control) or 100 Aekatperone (AKP) overnight in the presence of Rb$^+$. Before efflux measurements, cells were washed in an RbCl containing buffer lacking AKP for 30 min. Efflux was then measured for 30 min in a Ringer's solution±diazoxide (Diaz) at 200 μM. Note, diazoxide was included in Ringer's solution during the efflux assay but not during the overnight incubation. (**F**) Rb$^+$ efflux experiments showing overnight treatment with AKP enhances acute Diaz response in COSm6 cells expressing trafficking mutants. Each bar represents the mean ± SEM of at least three different biological repeats, with circles indicating individual data points, alpha = 0.05. *p<0.05, **p<0.01, ***p<0.001, ****p<0.0001 by one-way ANOVA with Dunnett's multiple comparisons test.

The online version of this article includes the following figure supplement(s) for figure 3:

**Figure supplement 1.** Acute inhibitory effect of Aekatperone purchased from Acme Bioscience, Inc.

further reduces the upper band of SUR1 trafficking mutants (*Martin et al., 2016*), diazoxide was not added during overnight Aekatperone treatment. Instead, COSm6 cells expressing either WT channels, SUR1$_{F27S}$, or SUR1$_{A30T}$ trafficking mutants were treated with Aekatperone (100 μM) for 16 hr, and diazoxide (200 M) was added only after Aekatperone washout during Rb$^+$ efflux measurements (see Methods; *Figure 3E*). As anticipated, untreated cells expressing the trafficking mutants exhibited a low response to diazoxide activation due to the reduced surface expression of K$_{ATP}$ channels, in contrast to cells expressing WT channels (*Figure 3F*). Significantly, mutant-expressing cells treated overnight with Aekatperone followed by subsequent drug removal showed much greater diazoxide response, achieving up to 80% of the WT response. These results further support that Aekatperone is a reversible inhibitor pharmacochaperone that rescues surface expression of CHI-causing K$_{ATP}$ trafficking mutants and allows functional recovery and diazoxide response of rescued mutant channels.

## Aekatperone shows selectivity toward SUR1-containing channels

In addition to pancreatic K$_{ATP}$ channels formed by SUR1 and Kir6.2, K$_{ATP}$ isoforms formed by assembly of SUR2A or SUR2B with either Kir6.1 or Kir6.2 are present in other tissues and cell types where they have critical physiological functions (*Shyng, 2022*). Drugs that preferentially target pancreatic K$_{ATP}$ channels would reduce undesired side effects arising from their effects on other K$_{ATP}$ isoforms. As K$_{ATP}$ inhibitor pharmacochaperones exert their effects by binding to a pocket in the SUR subunit, we assessed the acute inhibitory effects of Aekatperone on K$_{ATP}$ isoforms containing Kir6.2 and SUR2A or SUR2B, which are two major splice variants of SUR2, using both electrophysiology and Rb$^+$ efflux assays.

In electrophysiology experiments, we evaluated whether Aekatperone prevents the ability of SUR2A/Kir6.2 and SUR2B/Kir6.2 channels to respond to MgADP stimulation as seen in SUR1/Kir6.2 channels. Inside-out patch-clamp recordings showed that including 10 μM Aekatperone had no significant effects on MgADP stimulation of SUR2A/Kir6.2 nor SUR2B/Kir6.2 channels (*Figure 4A*), indicating SUR2A/Kir6.2 and SUR2B/Kir6.2 channels are relatively insensitive to Aekatperone compared to SUR1/Kir6.2 channels.

Rb$^+$ efflux experiments were then carried out to assess the effects of Aekatperone on SUR2-containing K$_{ATP}$ channel function in intact cells. COSm6 cells co-transfected with Kir6.2 and WT SUR2A or SUR2B cDNAs were subjected to Rb$^+$ efflux assays with or without Aekatperone (0–200 μM) in Ringer's solution containing metabolic inhibitors (see Methods). The results yielded dose-response curves

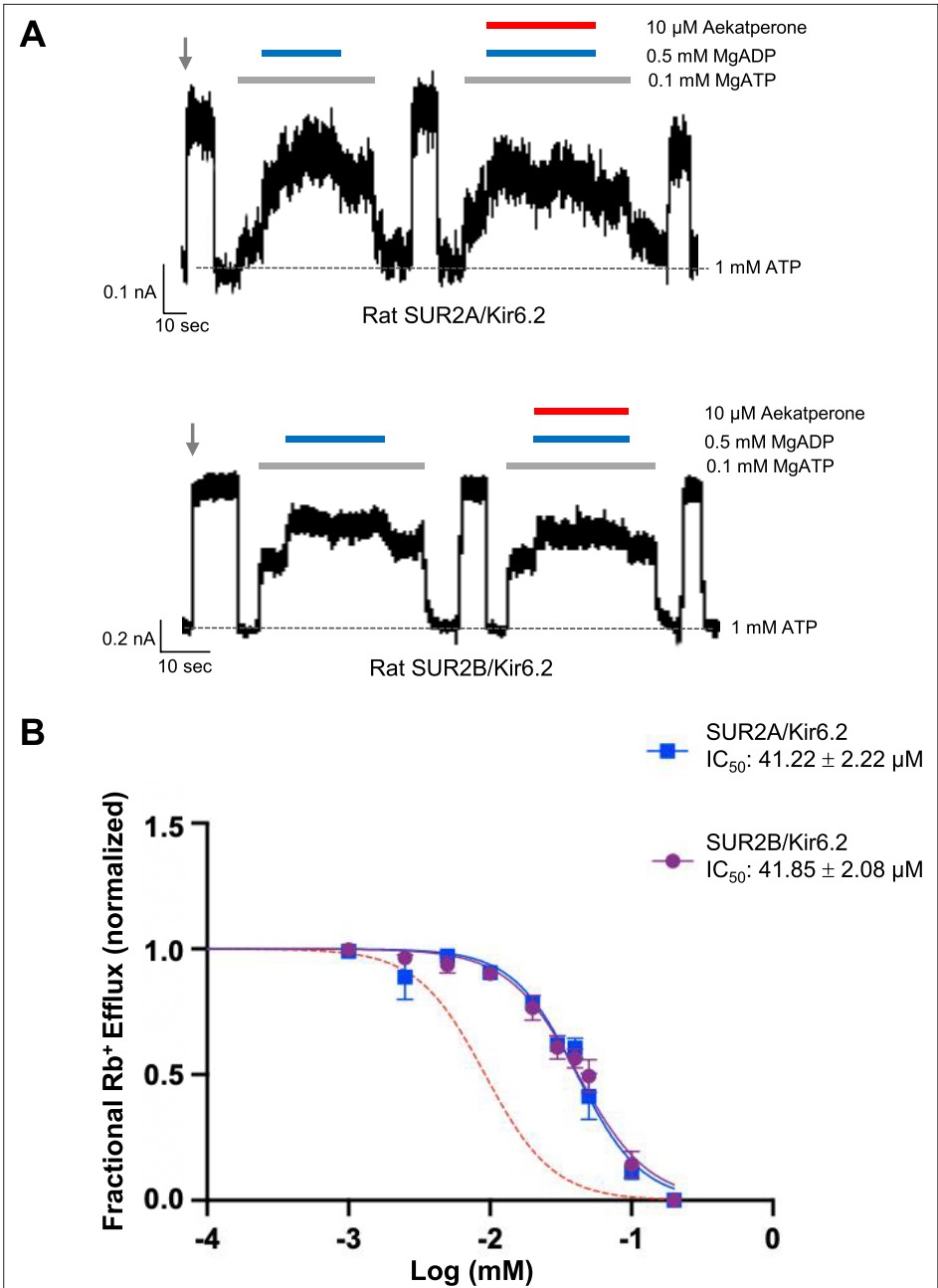

**Figure 4.** SUR2-containing $K_{ATP}$ channels are less sensitive to Aekatperone inhibitory effects. (**A**) Representative recordings from COSm6 cells expressing SUR2A/Kir6.2 (top) or SUR2B/Kir6.2 channels. Channels were exposed to K-INT solution upon patch excision (gray arrow) and exposed to solutions containing MgATP, MgADP, or Aekatperone as indicated by the bars above the recordings and the labels on the right. The patch was exposed to 1 mM ATP periodically to ensure the baseline has not shifted (gray dashed line). (**B**) Half-maximal inhibitory concentration ($IC_{50}$) of Aekatperone on SUR2A/Kir6.2 or SUR2B/Kir6.2 channels transiently expressed in COSm6 cells Kir6.2 using $Rb^+$ efflux assay. Data were fit with a Hill equation with variable slope using GraphPad Prism 10. The Aekatperone $IC_{50}$ is 41.22 µM±2.22 µM (SEM) for SUR2A/Kir6.2 channels and 41.85 µM±2.08 µM (SEM) for SUR2B/Kir6.2 channels. The error bar for each data point represents the SEM. Red dotted line represents the dose-response curve of Aekatperone on SUR1/Kir6.2 channels from *Figure 3B* for comparison.

showing an Aekatperone $IC_{50}$ of 41.22 µM±2.22 µM for SUR2A/Kir6.2 channels and 41.85 µM±2.08 µM for SUR2B/Kir6.2 channels (*Figure 4B*), which are more than fourfold higher than the $IC_{50}$ of 9.2 µM for SUR1/Kir6.2 channels (*Figure 3B*). These results further establish the preferential action of Aekatperone on SUR1- vs SUR2A/2B-containing $K_{ATP}$ channels.

## Aekatperone binding site revealed by cryoEM structure

Published cryoEM structures of $K_{ATP}$ channels bound to inhibitor pharmacochaperones, including GBC, RPG, and CBZ, have shown that these chemically diverse compounds share a common TMD binding pocket in SUR1 but have distinct chemical interactions with the channel (*Ding et al., 2019*; *Martin et al., 2017a*; *Martin et al., 2019*; *Martin et al., 2017b*; *Wu et al., 2018*). We anticipate Aekatperone to bind to the same pocket as the structure of this pocket was used for the initial virtual screening. Because AtomNet evaluates each compound based on an ensemble of possible poses, it remains uncertain how Aekatperone interacts with the channel. To directly visualize how Aekatperone interacts with the channel we determined the cryoEM structure of $K_{ATP}$ channels in the presence of 50 µM Aekatperone, using our published cryoEM sample preparation, imaging, and data processing workflow with specific modifications (*Driggers et al., 2024*; *Driggers and Shyng, 2021*; *Martin et al., 2017a*; *Martin et al., 2017b*) (see Methods and *Figure 5—figure supplement 1*).

3D reconstruction of cryoEM particles of channels incubated with Aekatperone yielded a map with an overall resolution of 4.1 Å. A model built from the map shows a closed Kir6.2 channel pore with a pore radius of ~0.75 Å at the helical bundle crossing residue F168 and an inward-facing SUR1 with NBDs separated, resembling other inhibitor-bound $K_{ATP}$ channel structures reported to date (*Figure 5A*). A clear nonprotein density consistent with the size of Aekatperone (MW: 399.52 g/mol) was observed in a TM pocket above NBD1 in the ABC core of SUR1 (*Figure 5B–D*). Assignment of this density to Aekatperone was supported by a side-by-side comparison with our published cryoEM map of the channel determined in the absence of any pharmacochaperone (EMD-26320; *Figure 5—figure supplement 2*). We have previously shown that this same pocket accommodates other inhibitor pharmacochaperones including GBC, RPG, and CBZ (*Martin et al., 2019*). Moreover, as in the structures of $K_{ATP}$ bound to GBC, RPG, or CBZ (*Martin et al., 2019*; *Sung et al., 2022*), we observed clear cryoEM density corresponding to the Kir6.2 distal N-terminal peptide domain (KNtp) in between the two TM bundles of the SUR1-ABC core, with the very N-terminus of Kir6.2 contacting the bound Aekatperone (*Figure 5A*). Stabilization of the KNtp-SUR1 interface promotes channel assembly, at the same time, it prevents Kir6.2 cytoplasmic domain from rotating to an open conformation (*Driggers et al., 2024*; *Martin et al., 2019*; *Sung et al., 2022*; *Wu et al., 2018*), which explains the dual chaperone and inhibitor activity of Aekatperone.

Aekatperone comprises a cyclohexane ring, a pyrazole, a sulfonamide moiety, a benzene ring, and an imidazole group (*Figure 1B*). The preferred binding conformation, which accounted for the physical chemical properties of the compound and its surrounding interacting protein residues as well as fit to the cryoEM density, is shown in *Figure 5B and C*. In this structural model, the cyclohexyl moiety of Aekatperone is aligned with S1238 on the cytoplasmic end of the binding pocket, and the imidazole group is oriented toward the extracellular end of the binding pocket, interacting with the distal part of KNtp. The pyrazole moiety has its two nitrogen atoms coordinated by T1242 and N1245, while the sulfonyl moiety is oriented toward R1246 and R1300, each interacts with one of the two oxygens of the sulfonyl group. Moreover, the benzene ring of Aekatperone forms a π–π stacking interaction with Y377 of SUR1, and a series of hydrophobic interactions contributed by TM helices from both TMD1 (TM6, 7) and TMD2 (TM16, 17) help stabilize the benzene and cyclohexyl groups. Interestingly, unlike GBC and RPG, the binding of both involves R306 of SUR1, Aekatperone does not reach far into the extracellular end of the pocket to interact with R306 (*Figure 5B and C*, *Figure 5—figure supplement 3*).

The atomic coordinates for Aekatperone fit well into the cryoEM density except for the imidazole group next to KNtp (*Figure 5B–D*). To investigate the dynamic behavior of Aekatperone in the SUR1 cavity, which may explain why the cryoEM density only partially covers the modeled structure of Aekatperone, we employed molecular dynamics (MD) simulations (*Figure 6—figure supplement 1*). During MD simulations of Aekatperone in its SUR1 binding pocket, Aekatperone remained within the pocket throughout the 500 ns simulation time in all five runs (a total of 2.5 µs) but exhibited some flexibility. To characterize the dynamics of Aekatperone, we divided its structure into five parts and

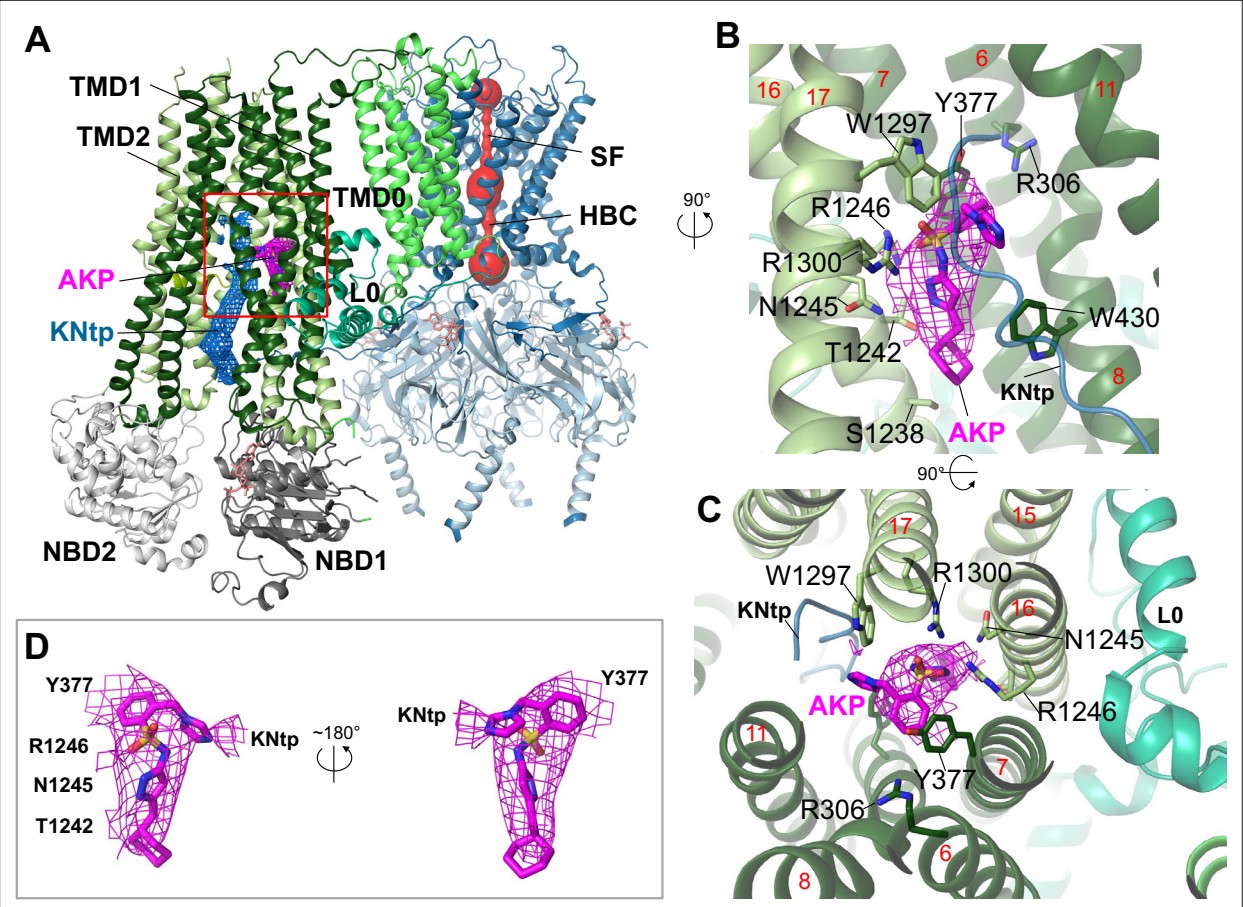

**Figure 5.** Structure of the K$_{ATP}$ channel in complex with Aekatperone. (**A**) Structural model of the pancreatic K$_{ATP}$ channel in complex with Aekatperone showing SUR1 in NBD-separated conformation and the Kir6.2 pore (red) constricted at the helical bundle crossing (HBC) and the selectivity filter (SF) gates. Aekatperone is shown as purple mesh (0.08 V contour) and N-terminal domain of Kir6.2 (KNtp) as blue mesh (0.08 V contour). Only one SUR1 subunit attached to Kir6.2 core is shown based on the focused refinement cryo-electron microscopy (cryoEM) map of the Kir6.2 tetramer and a single SUR1 subunit. SUR1 transmembrane domain (TMD) 1 and 2 are colored in dark green and light green, respectively. NBD, nucleotide binding domain; L0, L0 loop of SUR1. (**B, C**) Close-up side view (**B**) and top-down view (**C**) of the Aekatperone binding site. Aekatperone cryoEM density and model are shown in magenta and SUR1 residues that interact with Aekatperone are shown as sticks. KNtp is shown as a blue main chain peptide. Red numbers indicate the numbers of helices of SUR1. (**D**) Aekatperone cryoEM density map (0.08 V contour) and model fitting in two different views. Note at the contour shown, some surrounding density from interacting SUR1 and Kir6.2 is included.

The online version of this article includes the following figure supplement(s) for figure 5:

**Figure supplement 1.** Cryo-electron microscopy (cryoEM) data processing.

**Figure supplement 2.** Comparison of AKP-bound and AKP-free structures.

**Figure supplement 3.** Binding site model for (**A**) Aekatperone (AKP), (**B**) repaglinide (RPG, PDB ID 7TYS), (**C**) glibenclamide (GBC, PDB ID 7U24) bound to pancreatic K$_{ATP}$ channels, and (**D**) mitiglinide (MIT, PDB ID 7WIT) bound to a truncated SUR1 missing amino acids 2–208 in the absence of the Kir6.2 subunit.

analyzed the movement of each part individually. The positions of the centers of mass for each part, shown in corresponding colors in **Figure 6A**, indicate that while the positional distributions of most groups are relatively localized, group e, which corresponds to the imidazole ring, displays considerable conformational freedom. This is further illustrated in **Figure 6B**, where we calculated the standard deviation of the distances from their initial positions.

To some extent, the ligand's mobility is related to the flexibility of the amino acids within the surrounding pocket. The potential influence of conformational changes of SUR1 was reduced by aligning the trajectory on the Cα atoms of the pocket surrounding the ligand. However, side-chain mobility might still play a role. Therefore, we analyzed the ligand's surroundings and specifically investigated the frequency of the ligand remaining in close contact with the pocket amino acids.

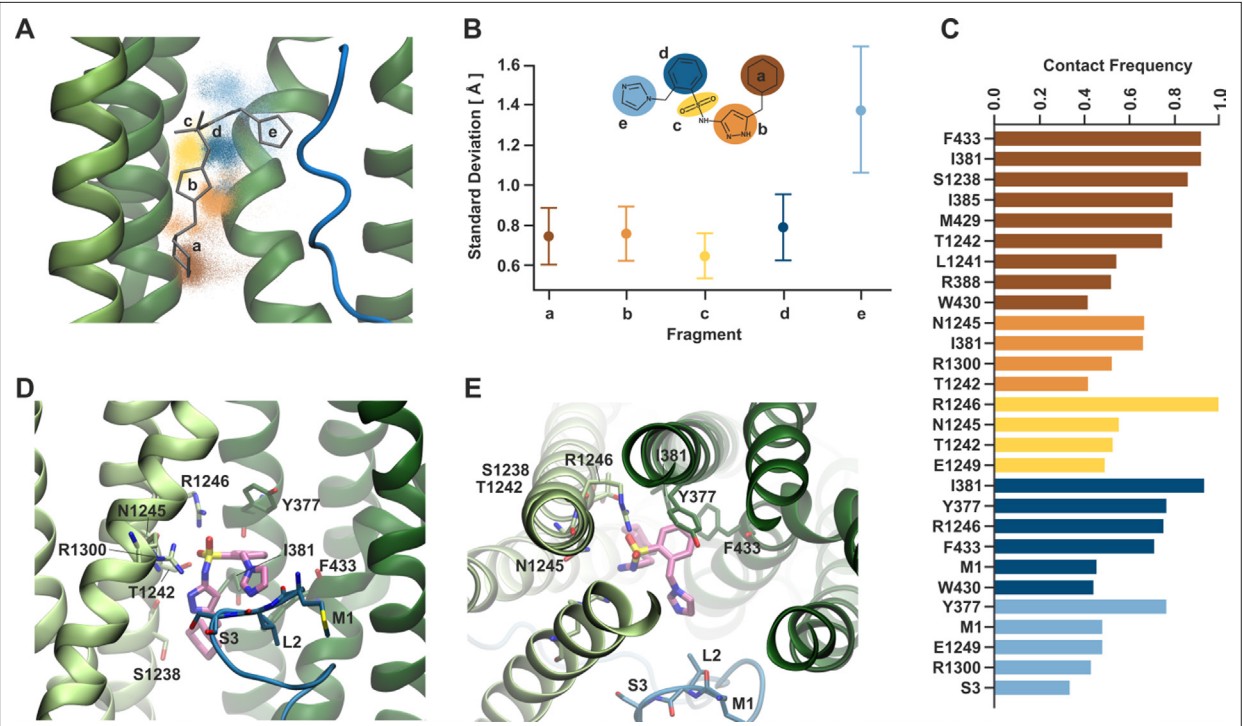

**Figure 6.** Dynamical behavior of Aekatperone in the SUR1 cavity investigated by molecular dynamics (MD) simulations. (**A**) Scatter of the positions of the centers of mass of individual fragments of Aekatperone during the simulation. The initial position is marked as sticks. Green helices are TMD1/2 of SUR1, and the blue fragment is KNtp. (**B**) Standard deviation of the positions of the centers of mass of individual fragments during the simulation. The fragments are color-coded throughout the figure. (**C**) Frequency of staying in close contact with SUR1 and KNtp residues during the simulation, divided by individual fragments. (**D, E**) An example snapshot from the simulation of Aekatperone in the pocket showing its interactions with SUR1 and KNtp residues - side view (**D**) and top view (**E**).

The online version of this article includes the following figure supplement(s) for figure 6:

**Figure supplement 1.** Properties of the system during the extended equilibration of the system (200 ns).

**Figure supplement 2.** Close contact frequency between first six residues of KNtp (labeled as KNt followed by residue number) and its surroundings in the Aekatperone (AKP) binding pocket.

*Figure 6C* shows the residues that stayed near the ligand for over 40% of the trajectory duration. Despite the ligand's mobility, some interactions remain at 80% or even 100%. These include the interaction of fragment a (brown) with F433, I381, and S1238, nearly 100% interaction of the sulfonyl group (yellow) with R1246, and slightly lesser interaction of group d (dark blue) with I381 and Y377. Y377 also participates to a slightly lower extent in binding fragment e. By sampling the available volume, the fragment e forms temporary interactions with KNtp amino acids (mainly M1 and S3), which also exhibit considerable conformational freedom within the pocket (*Figure 6—figure supplement 2*). An example snapshot from the simulation, showing Aekatperone in the SUR1 pocket and its interactions with SUR1 and KNtp amino acids, is depicted in *Figure 6D and E*, in a side and top views, respectively.

## Functional validation of the Aekatperone binding site model

To validate the above binding site model, we mutated a subset of Aekatperone-binding residues, including Y377, R1246, and R1300 to alanine (*Figure 7A*) and tested the effects of the mutation on channel response to the acute inhibitory effect of Aekatperone by Rb+ efflux assays. R306, which is predicted not to interact with Aekatperone, was also evaluated as a control. Consistent with our structural model, Y377A, R1246A, and R1300A all significantly decreased the dose-dependent inhibitory effects of Aekatperone (*Figure 7B*). In particular, the Y377A and R1300A mutations are detrimental to the ability of Aekatperone to inhibit the channel even at the highest concentration of Aekatperone (50 μM) tested, indicating a crucial role of these two residues. By contrast, R306A had little effects

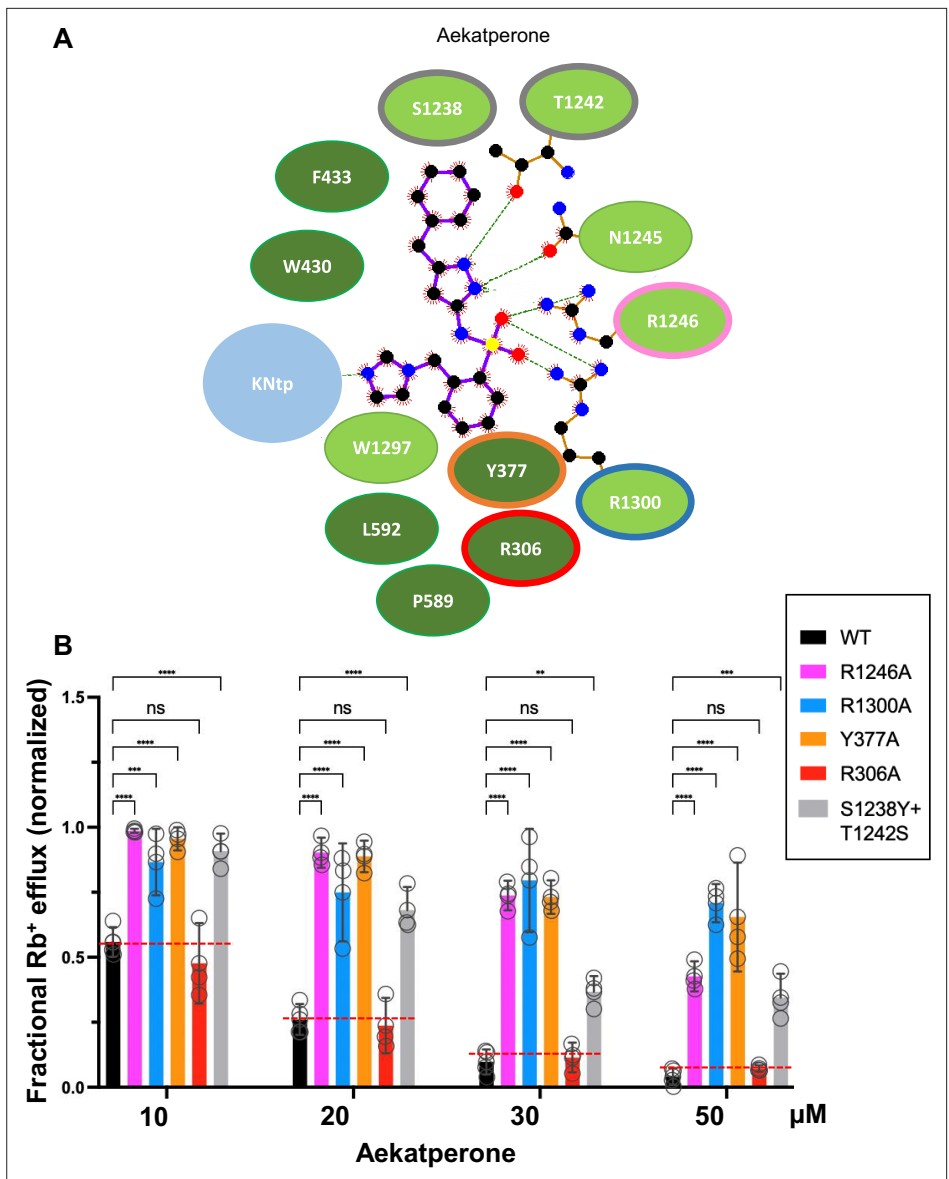

**Figure 7.** Mutagenesis studies supporting the Aekatperone binding site model derived from cryo-electron microscopy (cryoEM). (**A**) 2D Aekatperone binding site model showing chemical interactions between the compound and surrounding SUR1 residues and KNtp. (**B**) Rb$^+$ efflux experiments testing the effect of mutating select binding site residues on channel response to Aekatperone. Each bar is the mean and error bars represent the SEM of at least three independent experiments (individual data points shown as circles). Statistical significance is based on one-way ANOVA with Dunnett's post hoc multiple comparisons test, with alpha = 0.05. **$p<0.01$, ***$p<0.001$, ****$p<0.0001$.

on channel inhibition by Aekatperone, consistent with a lack of R306 involvement in Aekatperone binding.

In the Aekatperone binding site model, S1238 and T1242 of SUR1 lie next to the cyclohexyl and the pyrazole moieties (*Figure 5B and C*, *Figure 7A*, *Figure 5—figure supplement 3*). These two residues correspond to Y1205 and S1209 respectively in SUR2A and SUR2B. The substitution of the bulky tyrosine residue for serine at the SUR1-1238 position is expected to cause a steric interference with the bound drug, whereas serine substitution for threonine at the SUR1-1242 position requires serine to be in one of three possible rotomeric positions to allow interaction with the pyrozole group. We reasoned that these two amino acid substitutions in SUR2A/2B may account for the reduced Aekatperone inhibition seen in SUR2 isoforms compared to SUR1. To test this, we conducted Rb$^+$

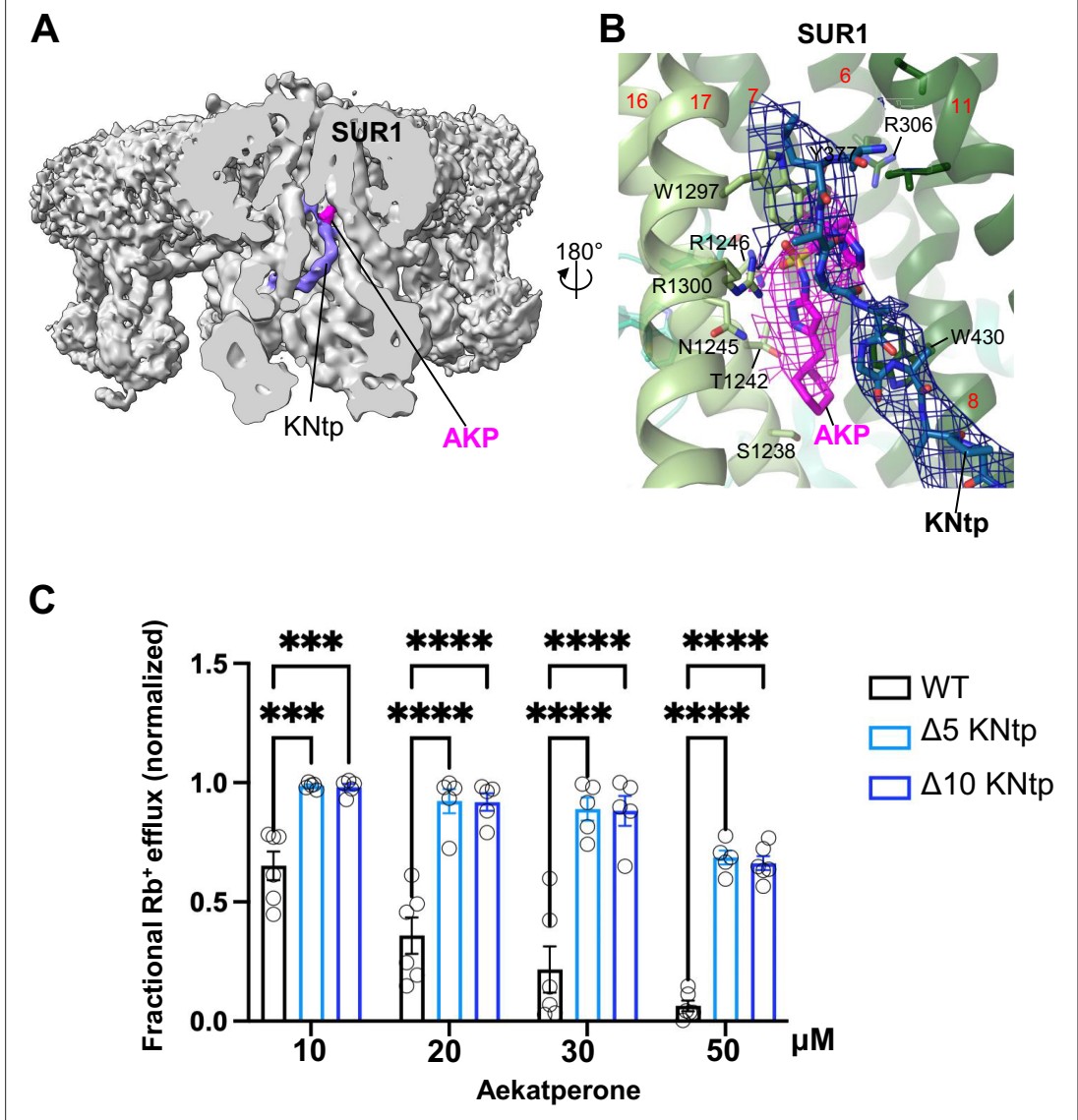

**Figure 8.** KNtp plays a role in Aekatperone-induced inhibitory effects on pancreatic $K_{ATP}$ channel. (**A**) Cryo-electron microscopy (cryoEM) density map of the pancreatic $K_{ATP}$ channel in complex with Aekatperone (AKP), with the SUR1 subunit in the front shown in vertical sliced view. KNtp is highlighted in blue and Aekaperone in magenta. (**B**) Close-up view of the AKP (magenta) and KNtp (blue) structures overlayed with corresponding cryoEM densities (at 0.08 V) showing the close proximity of the imidazole group of Aekatperone to KNtp. (**C**) Comparison of the effects of Aekatperone on COSm6 cells expressing either wild-type (WT) $K_{ATP}$ channels or channels with KNtp deletions of either 5 or 10 amino acids (Δ5 or Δ10 KNtp) using the $Rb^+$ efflux assay. $Rb^+$ efflux assays were performed in the presence of metabolic inhibitors with various concentrations of Aekatperone as indicated on the x-axis. The data were normalized against the fractional $Rb^+$ efflux of untreated cells expressing either WT or KNtp mutant channels with metabolic inhibition but without Aekatperone. Each bar is the mean and error bars represent the SEM of five to six independent experiments (individual data points shown as light black circles). Statistical significance is based on one-way ANOVA and Tukey's post hoc multiple comparisons test. Alpha = 0.05. **p<0.01, ***p<0.001, ****p<0.0001.

efflux assays in COSm6 cells expressing Kir6.2 and WT SUR1 or SUR1 with S1238Y and T1242S double mutation. The inhibitory effect of Aekatperone was significantly reduced by combined S1238Y and T1242S mutation compared to WT control at the 10–50 μM concentrations tested (*Figure 7B*). These findings further support the Aekatperone binding site model and offer a structural explanation for the lower sensitivity of the SUR2 channel isoform to the compound compared with the SUR1 isoform.

An additional structural element predicted to participate in Aekatperone binding based on our model is the distal N-terminus of Kir6.2 or KNtp (*Figure 5A*, *Figure 6C*, and *Figure 8A and B*). In particular, the imidazole group of Aekatperone is close to the cryoEM density corresponding to

the distal end of KNtp and may directly interact with KNtp residues (*Figure 5B and C*, *Figure 6C*, *Figure 8A and B*, *Figure 6—figure supplement 2*). To test the involvement of KNtp in channel inhibition by Aekatperone, we assessed the effects of deleting the N-terminal residues of Kir6.2 on channel response to 10, 20, 30, or 50 μM of the drug using $Rb^+$ efflux assays in COSm6 cells transiently co-transfected with $SUR1_{WT}$ along with $Kir6.2_{WT}$ or Kir6.2 lacking the distal 2–5 or 2–10 amino acids of Kir6.2 (referred to as $Kir6.2\Delta_{N5}$ or $Kir6.2\Delta_{N10}$). Both $Kir6.2\Delta_{N5}$ and $Kir6.2\Delta_{N10}$ significantly attenuated the inhibitory effect of Aekatperone at all four concentrations (*Figure 8C*). These results are in agreement with a role of KNtp in the binding of Aekatperone and/or transducing the effect of Aekatperone binding to channel closure. Taken together, the above functional data provides strong support for the Aekatperone binding site model derived from the cryoEM density map.

## Discussion

Here, we report the discovery and characterization of a new pancreatic-selective $K_{ATP}$ channel binder with dual pharmacochaperone and inhibitory effects. The compound, which we name Aekatperone, rescues $K_{ATP}$ channel trafficking mutants to the cell surface. While Aekatperone also inhibits channel activity, its inhibitory effect is readily reversible upon washout to allow functional recovery of rescued channels. These properties distinguish Aekatperone from existing high affinity $K_{ATP}$ inhibitory ligands with pharmacochaperone actions and make it a suitable candidate for combating CHI caused by $K_{ATP}$ trafficking defects. The cryoEM structure reveals that although Aekatperone sits in the same pocket that has been previously identified for high affinity inhibitors, it adopts a distinct binding position which may underlie its reduced apparent affinity and reversibility. Our study provides proof-of-principle for AI-assisted and structure-based $K_{ATP}$ channel drug discovery tailored to specific channel defects.

### AI-aided virtual screening in drug discovery

Compared to physical high-throughput screening of large compound libraries, computational virtual screening offers a faster and more cost-effective alternative approach to drug discovery. However, until recently computation-based drug development has been limited due to the scarcity of high-resolution protein structures especially for membrane proteins. The rapid rise in the number of high-resolution cryoEM structures of membrane proteins in the past decade and the mechanistic insights from these structures present an exciting opportunity for structure-based drug discovery. Taking advantage of cryoEM structures of the pancreatic $K_{ATP}$ channel bound to several known molecules with pharmacochaperone and inhibitory effects, we set out to search for additional small molecule binders for the pocket using the AI-aided virtual screening platform AtomNet, developed by Atomwise Inc. Out of the 96 top-ranking compounds, at least 3 with distinct scaffolds showed in the initial single-dose experimental screen promising pharmacochaperone and inhibitory effects on the channel to warrant follow-up studies. To our knowledge, this is the first structure-based virtual screening study conducted for $K_{ATP}$ channels. In contrast to other recent large-scale structure-based virtual screening studies involving docking of more than a billion compounds in target proteins such as GPCRs and CFTR (*Fink et al., 2022*; *Grotsch et al., 2024*; *Liu et al., 2024*), our AI-based virtual screening against the $K_{ATP}$ channel focused on ~2.5 million compounds in the smaller Enamine library, which can be readily purchased for experimental testing. The success in identifying several compounds validated by functional assays in our study illustrates the cost-effectiveness of this approach in drug discovery for $K_{ATP}$ channels. This approach has also been shown effective for other targets as recently documented in the Atomwise AIMS program (*Atomwise AIMS Program, 2024*). Thus, AI-aided structure-based virtual screening is emerging as a promising strategy for discovering novel therapeutics, a particularly important drug discovery development for rare diseases.

### Binding site comparison between Aekatperone and other $K_{ATP}$ inhibitors

Early structure-activity relationship studies of $K_{ATP}$ targeting drugs such as first and second generations of sulfonylureas, exemplified by tolbutamide and GBC respectively and glinides including RPG, along with comparative mutagenesis studies between SUR1 and SUR2 have led to a binding site model for $K_{ATP}$ inhibitors (*Quast et al., 2004*). In this model, the high affinity sulfonylurea GBC comprises two binding sites: site A which accommodates the sulfonylurea part of the molecule, and site B which

accommodates the benzamido end of the molecule, with the central phenyl ring shared between site A and site B (*Quast et al., 2004*). Tolbutamide which binds the channel with 1000-fold lower affinity than GBC and comprises only the phenyl ring and sulfonylurea group is thought to bind site A, whereas RPG which contains a phenyl ring and a benzamido moiety is thought to bind site B. Site A involves S1238 in SUR1, which is substituted by a tyrosine in SUR2. Mutation of SUR1 S1238 to Y abolishes tolbutamide binding, lowers GBC binding affinity, and renders GBC inhibition reversible, but has no effect on RPG binding. This model has been largely confirmed by recent cryoEM structures (*Driggers and Shyng, 2023*; *Wu et al., 2020*) (see *Figure 5—figure supplement 3*). According to this model, Aekatperone would be classified as a site A ligand. This is supported by structural data showing Aekatperone occupying the space corresponding to the site A part of GBC, and by mutagenesis data showing that the S1238Y-SUR1 mutation compromises channel inhibition by Aekatperone. Interestingly, whereas tolbutamide and Aekatperone, both site A ligands, block the channel with lower potencies ($IC_{50}$~23 µM and 9 µM, respectively) (*ElSheikh et al., 2024*) and are reversible, another site A ligand, mitiglinide (*Wang et al., 2022*) (see *Figure 5—figure supplement 3D*), is a much more potent and nearly irreversible inhibitor with an $IC_{50}$ in the low nM range (*Reimann et al., 2001*). Thus, site A ligands can exhibit a wide range of potencies. MD simulations of the Aekatperone binding site revealed dynamic interactions between the drug and amino acids in the channel from both subunits. Whether these dynamics underlie the lower affinity and reversibility of Aekatperone remains to be determined. Additional MD simulations and structure-function correlation studies may shed light on the mechanisms that dictate the apparent affinity and reversibility observed for the known inhibitor pharmacochaperones.

The Aekatperone-bound pancreatic $K_{ATP}$ structure also reveals that aside from S1238, another residue T1242, which is altered to a serine in SUR2, contributes to ligand binding and the ~4- to 5-fold higher selectivity of the ligand for SUR1 vs SUR2 channels. Moreover, our structural and functional data point to a role of Kir6.2 N-terminus in Aekatperone interaction and action. The expanding ligands and structure-function information will facilitate rational design of drugs with specific properties to meet a wide range of medical needs arising from $K_{ATP}$ dysfunction.

## Implications for CHI

As a rare disease, CHI has not been a major target for drug development. The majority of CHI with known genetic causes are due to loss-of-function mutations in the $K_{ATP}$ channel genes (*Hewat et al., 2022*; *Stanley, 2016*). Nearly half of CHI-$K_{ATP}$ mutations that have been studied prevent surface expression of the channel, rendering them unable to respond to the $K_{ATP}$ channel opener diazoxide, the mainstay medical treatment available for CHI (*ElSheikh and Shyng, 2023*). Unlike cystic fibrosis, in which 90% patients carry one or two copies of a highly prevalent trafficking mutation ΔF508 in CFTR (*Cutting, 2015*), there is no single highly prevalent mutation in the $K_{ATP}$ channel that underlies channel trafficking defects. However, we have shown that $K_{ATP}$ channel trafficking mutations are concentrated in SUR1-TMD0 (*ElSheikh and Shyng, 2023*; *Martin et al., 2020*). $K_{ATP}$ inhibitors, including tolbutamide, GBC, RPG, and CBZ, as well as Aekatperone described here rescue SUR1-TMD0 trafficking mutants to the cell surface. Thus, inhibitor pharmacochaperones would have general applications to a significant number of CHI patients.

An additional challenge for the pharmacochaperone therapeutic approach is the inhibitory nature of the $K_{ATP}$ pharmacochaperones. CryoEM structures suggest that $K_{ATP}$ inhibitors bind to the SUR1 ABC core domain and stabilizes SUR1-Kir6.2 interactions to promote mutant channel assembly (*Martin et al., 2019*). However, stabilization of this interface also restricts the movement of the Kir6.2 cytoplasmic domain toward the open conformation (*Driggers et al., 2024*; *Martin et al., 2019*; *Sung et al., 2022*; *Wu et al., 2018*). As such, the principal challenge in the clinical use of pharmacological chaperones for patients with CHI lies in balancing channel activity enhancement by boosting surface expression with activity inhibition via gating. GBC and RPG bind $K_{ATP}$ with such high affinity that they prevent functional recovery of rescued channels. CBZ, while more reversible, has multiple known ion channel targets including $Na_v$, $Ca_v$, and NMDA receptors (*Ambrósio et al., 1999*; *Gambeta et al., 2020*; *Matsumoto et al., 2015*; *Tikhonov and Zhorov, 2017*). Tolbutamide would be suitable due to its low binding affinity and reversibility; however, it has long been taken off market due to concerns over its adverse cardiovascular effects (*Ashcroft, 2005*; *Schwartz and Meinert, 2004*), possibly by interacting with hERG channels (*Garrido et al., 2020*). Thus, new $K_{ATP}$ pharmacochaperones are needed.

Aekatperone is a promising lead as it fits the profile of a reversible inhibitor pharmacochaperone and shows little effects on SUR2 $K_{ATP}$ isoforms in the concentration range for its pharmacochaperone effects on pancreatic $K_{ATP}$ channels. Significantly, we showed that upon Aekatperone overnight rescue and quick washout, diazoxide greatly enhanced efflux in cells transfected with trafficking mutant channels to nearly 80% of that observed for WT channels. A dosing regimen incorporating diazoxide and a reversible inhibitor pharmacochaperone such as Aekatperone to harness the channel potentiating effect of diazoxide would further enhance the therapeutic effect of the pharmacochaperone.

### Limitations and future directions

We show that Aekatperone is ~4- to 5-fold more selective for SUR1-containing pancreatic $K_{ATP}$ over SUR2A/B containing $K_{ATP}$ based on functional studies. The selectivity suggests clinical use may involve fewer side effects due to isoform cross-reactivity. However, we have not tested the compound against other protein targets (see discussion on CBZ and tolbutamide above), therefore potential off-target effects remain. In this regard, it is interesting that nateglinide, a antidiabetic glinide drug which would be predicted to bind the same pocket as Aekatperone, has been shown to be a low micromolar agonist for the human itch GPCR MRGPRX4 (*Cao et al., 2021*). Clearly, before Aekatperone can be applied to CHI treatment, it needs to be rigorously evaluated for potential adverse cardiovascular and neurological effects.

Compared to ultra-large library docking of over a billion make-on-demand compounds, our focused approach on a commercially available smaller compound library may miss potential scaffolds that will bind to our target and yield the desired biological effects. We envision the AI-based screening can be scaled up to cover even more diverse molecules.

Finally, we demonstrate surface expression and functional rescue of $K_{ATP}$ channel trafficking mutants by Aekatperone in a cell culture experimental system. Translation of this finding to clinical applications will require many more studies using relevant cell and animal models. Nonetheless, our study offers a framework for structure- and mechanism-based drug development for $K_{ATP}$ channels.

## Methods

### Virtual screening using the AtomNet platform

The virtual screening was carried out using AtomNet, within the framework of Atomwise's academic collaboration program known as Artificial Intelligence Molecular Screen (AIMS). AtomNet is a deep neural network for structure-based drug design and discovery (*Gniewek et al., 2021*; *Wallach et al., 2015*). An Enamine in-stock library of ~2.5 million commercially available compounds was first filtered using the Eli Lilly medicinal chemistry filters to remove potential false positives from the group, including aggregators, autofluorescence, and pan-assay interference compounds (*Baell and Holloway, 2010*; *Bruns and Watson, 2012*). The filtered library was then virtually screened using the cryoEM structure of the $K_{ATP}$ channel bound to RPG (PDB ID: 6PZ9, ~3.65 Å)(*Martin et al., 2019*), targeting the binding site within the central cavity of the SUR1 subunit of the $K_{ATP}$ channel (*Figure 1A*). Molecules with greater than 0.5 Tanimoto similarity in ECFP4 space to any known binders of the target and its homologs within 70% sequence identity were excluded from the virtual screen to increase the possibility of identifying novel hits. For each small molecule, we generated an ensemble of binding poses within the target site. These poses were then scored utilizing the AtomNet technology, enabling ranking of molecules based on their predicted affinity. To ensure diversity in the selected compounds and to minimize the selection of structurally similar scaffolds, the top 30,000 molecules were clustered using the Butina algorithm (*Butina, 1999*) with a Tanimoto similarity cutoff of 0.35 in ECFP4 space for the final compound selection list. Furthermore, we applied filters to assess oral availability and to eliminate compounds containing undesirable substructures and molecular properties known as toxicophores from the list. Detailed description of the AIMS screening protocol can be found in recent literature (*Atomwise AIMS Program, 2024*). The top-ranking compounds predicted to exhibit high affinity binding were subsequently purchased for experimental testing.

### Compounds for functional screening

Top-ranking 96 compounds (in powder form of ~2–5 mg) were purchased from Enamine (Kyiv, Ukraine). Because of the small quantity of powder compounds used for reconstitution, the concentrations were

not highly accurate. Upon determining that Aekatperone was the compound to be characterized in detail, a larger quantity of Aekatperone was purchased from Enamine. Thus, a small error range between the dose-response from the initial functional screening and subsequent experiments exists. To ensure that the Aekatperone biological effects we observed on $K_{ATP}$ channels were not due to minor contaminants, we also purchased Aekatperone synthesized from a different commercial source, Acme Bioscience, Inc (Palo Alto, CA, USA).

### Expression constructs

Hamster SUR1 was tagged with a FLAG epitope (DYKDDDDK) at the extracellular N-terminus (referred to as FLAG-SUR1) and the cDNA cloned into the pECE vector. We have shown in previous studies that these epitope tags do not affect the trafficking or function of the channel (*Cartier et al., 2001*; *Lin et al., 2005*). WT rat Kir6.2 cDNA was cloned into pcDNA1/Amp, as described previously (*Cartier et al., 2001*). Point mutations to FLAG-SUR1 were introduced using the QuikChange site-directed mutagenesis kit (Agilent Technologies). All mutations were confirmed by DNA sequencing, and mutant clones from two independent PCRs were analyzed in all experiments to avoid false results caused by undesired mutations introduced by PCR. For experiments involving human $K_{ATP}$, WT or mutant human SUR1 cDNA and WT human Kir6.2 cDNA were cloned into pCMV6b and pcDNA3.1(+) vectors, respectively. Also, WT rat SUR2A and WT rat SUR2B that were cloned into pcMV6 were used in experiments that aimed to test isoform specificity. Construction of adeno-viruses carrying $K_{ATP}$ subunits cDNAs was as described previously (*Lin et al., 2008*). The hamster FLAG-SUR1$_{F27S}$ recombinant adenovirus was made using a modified pShuttle plasmid (AdEasy kit, Agilent Technologies) containing a tetracycline-inducible promoter and requires co-infection of a virus carrying the cDNA encoding a tetracycline-inhibited transactivator (tTA) for expression. Recombinant viruses were amplified in HEK293 cells and purified according to the manufacturer's instructions.

### Cell lines

COSm6 cells (RRID:CVCL_8561), a Green monkey (*Cercopithecus aethiops*) kidney cell line (*Horowitz et al., 1983*), was used for protein expression for immunoblotting, electrophysiology, and Rb$^+$ efflux experiments. INS-1 clone 832/13 (RRID:CVCL_7226) (*Hohmeier et al., 2000*), a rat (*Rattus norvegicus*) insulinoma cell line (referred to as INS-1), was used for protein expression for compound screening and cryoEM studies. COSm6 cells were authenticated by RT-PCR of the mRNA for α-tubulin followed by sequencing to confirm Green monkey species identity. INS-1 cells were authenticated by RT-PCR of the mRNAs for SUR1 and Kir6.2 followed by sequencing to confirm rat identity. Both cell lines were routinely (~every 6 months) tested to exclude mycoplasma contamination throughout the course of the study.

### Transduction of INS-1 cells with recombinant adenoviruses for experimental screening of the top 96 scoring compounds from the virtual screening

INS-1 cells were plated in 15 cm plates and cultured for 24 hr in RPMI 1640 with 11.1 mM D-glucose (Invitrogen) supplemented with 10% fetal bovine serum, 100 units/ml penicillin, 100 µg/ml strep-tomycin, 10 mM HEPES, 2 mM glutamine, 1 mM sodium pyruvate, and 50 µM β-mercaptoethanol (*Hohmeier et al., 2000*). Cells at ⚓70% confluency were washed once with Dulbecco's Phosphate Buffered Saline (Sigma-Aldrich) and then incubated for 2 hr at 37°C in Opti-MEM reduced serum medium (Gibco) and a mixture of viruses including the tTA virus, the tTA-regulated hamster FLAG-SUR1$_{F27S}$ virus, and the rat Kir6.2$_{WT}$ virus as described previously (*Driggers and Shyng, 2021*; *Lin et al., 2005*; *Lin et al., 2008*). The multiplicity of infection (MOI) for each virus was determined empirically and the amount of virus needed to achieve the necessary MOI was calculated as follows: (number of cells in dish or well × desired MOI)/titer of the virus stock (pfu/ml) (*Driggers and Shyng, 2021*). After 120 min, INS-1 growth medium was added and the cells were incubated at 37°C for an additional 16–17 hr in RPMI 1640 with 10% FCS containing either DMSO, GBC, or each of the top scoring (96) compounds before harvesting for immunoblotting.

## Transfection of COSm6 cells with recombinant DNA

COSm6 cells plated in six-well tissue culture plates at ~70% confluency were transfected with 1.2 µg of WT or mutant SUR1, SUR2A, or SUR2B and 1.2 µg of Kir6.2 per well using FuGENE 6 (Promega) according to the manufacturer's directions. It is essential to initially mix the plasmids in Opti-MEM in a separate tube before adding the FuGENE to ensure the uptake of both SUR and Kir6.2 plasmids required for $K_{ATP}$ channel expression by the cell.

## Immunoblotting

INS-1 cells transduced with recombinant adenoviruses or COSm6 cells transfected with SUR1 and Kir6.2 cDNAs were lysed in lysis buffer (50 mM Tris·HCl, pH 7.0, 150 mM NaCl, and 1% Triton X-100, with cOmplete Mini Protease inhibitor from Roche) on ice for 30 min. Cell lysate was centrifuged at 16,000 × $g$ for 5 min at 4°C, and an aliquot of the supernatant was run on SDS-PAGE and transferred to nitrocellulose membrane. The membrane was probed with a rabbit anti-SUR1 serum raised against a C-terminal peptide of SUR1 (KDSVFASFVRADK) (*Yan et al., 2007*), followed by incubation with horseradish peroxidase-conjugated secondary antibodies (Amersham Biosciences), and visualized by enhanced chemiluminescence (Super Signal West Femto; Pierce). Tubulin was also probed and served as a loading control.

## Patch-clamp recording

COSm6 cells were transfected with SUR1, SUR2A, or SUR2B and Kir6.2 cDNAs along with cDNA encoding the green fluorescent protein to identify transfected cells. Cells were plated onto glass coverslips 24 hr after transfection and recordings made in the following 2 days. To test the acute effect of the drug, cells were subjected to inside-out patch voltage-clamp recording in solutions containing 1 mM ATP, 0.1 mM ATP, or 0.1 mM ATP plus 0.5 mM ADP with or without the drug Aekatperone (10 µM), MgCl$_2$ was added such that the free Mg$^{2+}$ concentration of the solutions was ~1 mM (*Shyng and Nichols, 1998*). Micropipettes were pulled from non-heparinized Kimble glass (Fisher Scientific) on a horizontal puller (Sutter Instrument, Novato, CA) with resistance typically ~1–2 MΩ. The bath (intracellular) and pipette (extracellular) solutions into which membranes were excised were K-INT (140 mM KCl, 10 mM K-HEPES) plus 1 mM K-EDTA, pH 7.3. ATP and ADP were added as potassium salts. Inside-out patches of cells were voltage-clamped with an Axopatch 1D amplifier (Axon Instruments). Recording was performed at room temperature (*Devaraneni et al., 2015*) and currents were measured at a membrane potential of –50 mV. Inward currents in all figures are shown as upward deflections. Data were analyzed using pCLAMP10 software (Axon Instrument). Offline analysis was performed using Clampfit and GraphPad. Data were presented as mean ± standard error of the mean (SEM).

## Immunofluorescence staining

COSm6 cells transfected with WT Kir6.2 and WT, A30T, or F27S FLAG-SUR1 were plated on coverslips 1 day before the experiment and treated overnight with DMSO or Aekatperone. To label channels expressed at the cell surface, cells were washed once with ice-cold phosphate buffered saline (PBS): 137 mM NaCl, 2.7 mM KCl, 10 mM Na$_2$HPO$_4$, 1.8 mM KH$_2$PO$_4$, pH 7.4 and then incubated with anti-FLAG M2 antibody (Sigma-Aldrich; 10 µg/ml in Opti-MEM plus 0.1% BSA) for 1 hr at 4°C. Cells were washed with ice-cold PBS and fixed with ice-cold 4% paraformaldehyde for 10 min on ice, and washed three times with cold PBS. Fixed cells were then incubated in blocking buffer (PBS+2% BSA+1% normal goat serum) for 1 hr, followed by incubation with Alexa Fluor 546-conjugated goat anti-mouse secondary antibody (Invitrogen; 1:300 dilution in blocking buffer) for 1 hr at room temperature. For staining of total cellular FLAG-SUR1, the cells were fixed with ice-cold 4% paraformaldehyde for 10 min on ice before incubation with anti-FLAG antibody followed by Alexa Fluor 546-conjugated goat anti-mouse secondary antibody. The cells were then washed twice with PBS and the coverslips were mounted on microscope slides using Vectashield Mounting Medium for Fluorescence with DAPI. Cells were viewed using an Olympus confocal microscope.

## Rb$^+$ efflux assay

COSm6 cells were transiently transfected with various combinations of WT or mutant SUR1 and Kir6.2 cDNAs. Untransfected cells were included as background control. Cells were cultured in medium

containing 5.4 mM RbCl overnight. The next day, cells were washed quickly twice in PBS with no RbCl. For experiments testing the acute inhibitory effects of Aekatpeone (*Figure 2—figure supplement 2*, *Figure 3A*, *Figure 4B*, *Figure 3—figure supplement 1*), $Rb^+$ efflux was measured by incubating cells in Ringer's solution (5.4 mM KCl, 150 mM NaCl, 1 mM $MgCl_2$, 0.8 mM $NaH_2PO_4$, 2 mM $CaCl_2$, 25 mM HEPES, pH 7.2) with metabolic inhibitors (2.5 µg/ml oligomycin and 1 mM 2-deoxy-D-glucose) and combined with the drug, for 30 min at 37°C (*ElSheikh et al., 2024*).

For experiments testing the pharmacochaperone effects of Aekatperone (*Figure 3C and D*), the drug was added to the RbCl-containing media overnight and an additional wash step in RbCl wash buffer (5.4 mM RbCl, 150 mM NaCl, 1 mM $MgCl_2$, 0.8 mM $NaH_2PO_4$, 2 mM $CaCl_2$, 25 mM HEPES, pH 7.2) for 30 min at 37°C was included to remove Aekatperone prior to the efflux assay. $Rb^+$ efflux was measured by incubating cells in Ringer's solution with only metabolic inhibitors without Aekatperone (*ElSheikh et al., 2024*). Similarly, for experiments evaluating the role of Aekatpeone in enhancing diazoxide response (*Figure 3E and F*), Aekatperone was included in the RbCl-containing medium overnight with an additional drug washout step, and the efflux was conducted in Ringer's solution containing 200 µM diazoxide.

For all efflux experiments, efflux solution was collected at the end of a 30 min incubation period and cells were lysed in Ringer's solution plus 1% Triton X-100. $Rb^+$ concentrations in both the efflux solution and cell lysate were measured using an Atomic Adsorption Instrument Ion Channel Reader (ICR) 8100 from Aurora Biomed. Fractional $Rb^+$ efflux was calculated by dividing $Rb^+$ in the efflux solution over total $Rb^+$ in the efflux solution and cell lysate. Fractional $Rb^+$ efflux from untransfected COSm6 cells was subtracted from other experimental readings. The data were normalized to the fractional $Rb^+$ efflux of cells expressing WT channels treated with metabolic inhibitors without any drugs, unless specified otherwise. For each experiment, technical duplicates were included, and the average value was used as the experimental result. At least three separate transfections were performed for each experimental condition as biological repeats, as detailed in the figure legends (number of biological repeats represented as small circles on the graphs). Data are presented as mean ± SEM.

## Protein expression and purification for cryoEM

$K_{ATP}$ channels were expressed and purified as described previously (*Martin et al., 2017b*). Briefly, eight 15 cm tissue culture plates of INS-1 cells clone 832/13 were transduced with the tTA adenovirus and recombinant adenoviruses carrying hamster FLAG-SUR1 and rat Kir6.2 cDNAs (*Driggers and Shyng, 2021*). After growing for approximately 48 hr post-infection, cells were washed with PBS and harvested by scraping. Cell pellets expressing FLAG-SUR1 and Kir6.2 were frozen in liquid nitrogen and stored at –80°C until purification.

For purification, cells were resuspended in hypotonic buffer (15 mM KCl, 10 mM HEPES pH 7.5) with protease inhibitor, PI (cOmplete Mini Protease inhibitor, Roche, Basel, Switzerland), and 1.0 mM ATP and lysed by Dounce homogenization. The total membrane fraction was prepared, and membranes were resuspended in Membrane Solubilization Buffer (MSB; 200 mM NaCl, 100 mM KCl, 50 mM HEPES pH 7.5) containing PI, 1 mM ATP, 4% wt/vol trehalose, and 50 µM Aekatperone and solubilized with 0.5% Digitonin. The soluble fraction was incubated with anti-FLAG M2 affinity agarose for 12 hr, washed three times with MSB+0.5% Digitonin+PI+ATP+Aekatperone solution, and eluted for 60 min at 4°C with 1.0 mL of a solution like the final wash but containing 0.25 mg/ml FLAG peptide. The eluted sample was loaded onto a Superose 6 10/300 column run by an ÄKTA FPLC (Cytiva Life Sciences), and 0.5 ml fractions were collected. Fractions corresponding to the hetero-octameric $K_{ATP}$ channels were collected and concentrated to a volume of ~60 µl and used immediately for cryo grid preparation.

## Sample preparation and data acquisition for cryoEM analysis

Using a Vitrobot Mark III (FEI), a 3 µl aliquot of purified and concentrated $K_{ATP}$ channel was loaded onto Lacey Carbon grids that had been glow-discharged for 60 s at 15 mA with a Pelco EasyGlow. The grid was blotted for 2 s (blot force –4; 100% humidity at 4°C) and cryo-plunged into liquid ethane cooled by liquid nitrogen to flash vitrify the sample.

Single-particle cryoEM data was collected on a Titan Krios 300 kV cryo-electron microscope (Thermo Fisher Scientific) in the Pacific Northwest CryoEM Center (PNCC), with a multi-shot strategy using beam shift to collect 19 movies per stage shift, assisted by the automated acquisition program

SerialEM. Movies were recorded on the Gatan K3 Summit direct-electron detector in super-resolution mode with Fringe-free imaging, post-GIF (20 eV window), at ×81,000 magnification (calibrated image pixel size of 1.0515 Å; super-resolution pixel size 0.52575 Å); nominal defocus was varied between –1.0 and –2.5 μm across the dataset. The dose rate was kept around 17 e$^-$/Å$^2$/s, with a frame rate of 23 frames/s, and 74 frames in each movie (i.e. 3.2 s exposure time/movie), which gave a total dose of approximately 55 e$^-$/Å$^2$. Two grids that were prepared in the same session using the same protein preparation were used for data collection, and from these two grids 21,012 movies were recorded. As Lacey carbon grids do not have a regular array of holes, many of the movies contain a large fraction of carbon support in the images.

## Image processing and model building

Super-resolution dose-fractionated movies were gain-normalized by inverting the gain reference in Y and rotating upside down, corrected for beam-induced motion, aligned, and dose-compensated using Patch-Motion Correction in cryoSPARC2 (*Punjani et al., 2017*) and binned with a Fourier-crop factor of ½. Parameters for the contrast transfer function (CTF) were estimated from the aligned frame sums using Patch-CTF Estimation in cryoSPARC2. The resulting 21,012 dose-weighted motion-corrected summed micrographs were used for subsequent cryoEM image processing. Particles were picked automatically using template-based picking in cryoSPARC2 based on 2D classes obtained from cryoSPARC live during data collection. Incorrect particle picks containing carbon edges, or features on the support carbon, were cleaned by multiple rounds of 2D classification in cryoSPARC2. The resulting stack contained 78,490 particles, which were extracted using a 512$^2$ pixel box and were used for ab initio reconstruction in C1 followed by homogeneous refinement in C1, which gave a gold-standard FSC resolution cutoff (GSFSC) (*Rosenthal and Henderson, 2003*) of about 6.3 Å. Homogeneous refinement with C4 symmetry imposed gave a GSFSC cutoff of 4.8 Å resolution. Reconstruction from all particles shows an NBD-separated SUR1 with a closed Kir6.2 pore and cryoEM density at the putative Aekatperone binding site.

To assess conformational heterogeneity within the analyzed particles and to improve the density for Aekatperone, particles were subjected to fourfold symmetry expansion at the C4 axis. A mask covering the Kir6.2 tetramer, TMD0 of the four SUR1 subunits, plus a large volume surrounding one of the full SUR1 subunits, to have a large region covering all possible SUR1 locations (*Figure 5—figure supplement 1A*), was generated using Chimera and used as a focused map to conduct 3D classification of these 313,960 symmetry-expanded particles without particle alignment in cryoSPARC2. This yielded three dominant classes all with closed Kir6.2 pores and separated SUR1-NBDs that differed in the position of the cytoplasmic domain (CTD) of Kir6.2 (*Figure 5—figure supplement 1A*). A class of 125,416 symmetry-expanded particles in the CTD-up conformation gave a 4.1 Å resolution reconstruction (masked), and the cryoEM density map from this reconstruction was used to build a model of the K$_{ATP}$ channel bound to Aekatperone. In this reconstruction of the full K$_{ATP}$ channel particle, there is sufficient density to define helical positions and domain positions that are similar to RPG- or GBC-bound structures previously published. The TM helices have higher local resolution than the dynamic NBDs and loop regions (*Figure 5—figure supplement 1B*).

To create an initial structural model, PDB ID 7TYS had ligands removed and then was fit into the reconstructed density for the full K$_{ATP}$ channel particle using Chimera, and then refined in Phenix as separate rigid bodies corresponding to TMD (32–171) and CTD (172–352) of Kir6.2 and TMD0/L0 (1–284), TMD1 (285–614), NBD1 (615–928), NBD1-TMD2-linker (992–999), TMD2 (1000–1319), and NBD2 (1320–1582) of SUR1. The model containing Aekatperone was then built manually using *Coot*. The resulting model was further refined using *Coot* and *Phenix* iteratively until the statistics and fitting were satisfactory (*Supplementary file 2*, *Figure 5—figure supplement 1C*). All cryoEM structure figures were produced with UCSF Chimera, ChimeraX, and PyMol (http://www.pymol.org). Pore radius calculations were performed with HOLE implemented in *Coot*.

## MD simulations

We used Gromacs 2023 for MD simulations, based on the Aekatperone-bound K$_{ATP}$ structural model with intrinsically disordered regions reconstructed as in our previous work (*Walczewska-Szewc and Nowak, 2023*). The system, containing four Kir6.2 and one SUR1, was embedded in a 1-palmitoyl-2-oleoyl-phosphatidylcholine lipid bilayer with water (TIP3) and 0.2 M KCl using CHARMM-GUI (*Jo*

*et al., 2008*; *Wu et al., 2014*). We applied the CHARMM36m force field for the whole system (*Huang et al., 2017*). A Swiss param server was used to generate parameters for Aekatperone (*Bugnon et al., 2023*; *Zoete et al., 2011*). Short-range Coulomb and van der Waals interactions had 1.0 nm cutoffs, with long-range electrostatics managed by the particle-mesh Ewald method. Bonds were restrained using LINCS. Schrodinger software predicted pKa values and ligand protonation state (*Hess, 2008*). Energy minimization used the steepest descent algorithm. A 200 ns NVT equilibration with position restraints on backbone and ligand atoms (fc = 1000 kJ/mol/nm²) allows the solvent to adapt to the pore and relaxed IDRs (*Figure 6—figure supplement 1*). Five 500 ns production runs were in an NPT ensemble with a velocity-rescale thermostat at 309 K ($\tau$ =0.1 ps) and a Parrinello-Rahman barostat at 1 bar ($\tau$ =2 ps) (*Bussi et al., 2007*; *Parrinello and Rahman, 1981*).

The stability of Aekatperone binding was analyzed using custom Python3 scripts (https://doi.org/10.18150/PSA6NC) based on the MDAnalysis library (*Michaud-Agrawal et al., 2011*). Trajectories were aligned on helices 11, 12, and 14 of the TMD1 and TMD2 domains of SUR1, which form the Aekatperone binding site. The space sampled by the ligand during the simulation was visualized by tracking the center of mass positions of groups a, b, c, d, and e (*Figure 6*). This allowed us to assess the binding stability and conformational freedom of the entire ligand and each part separately. Key residues involved in ligand binding were identified based on their frequency of remaining in close contact (within 3.5 Å) with the ligand throughout the simulation. Residues that remained in close contact for at least 30% of the simulation time in at least three out of five trajectories were included. MD simulation figures were generated using VMD 1.9.3.

## Statistics

Data are presented as mean ± SEM. Differences were tested using Student's t-test when comparing two groups or one-way analysis of variance (ANOVA) when comparing three or more groups with Tukey's or Dunnett's post hoc tests for multiple comparisons in Graph Pad Prism 10 as indicated in figure legends. Differences were assumed to be significant if $p \leq 0.05$.

## Acknowledgements

A portion of this research was supported by NIH grant U24GM129547 (proposal 51311 to SLS) and performed at the Pacific Northwest Cryo-EM Center (PNCC) at Oregon Health & Science University and accessed through EMSL (grid.436923.9), a DOE Office of Science User Facility sponsored by the Office of Biological and Environmental Research, with special thanks to Dr. Nancy Meyer for help with cryoEM data collection. We acknowledge support by the National Institutes of Health grant R01GM145784 (to SLS) and the Egyptian government predoctoral scholarship GM 1109 (to AE). We also acknowledge Polish high-performance computing infrastructure PLGrid for awarding this project access to the LUMI supercomputer, owned by the EuroHPC Joint Undertaking, hosted by CSC (Finland) and the LUMI consortium through PLL/2023/04/016512 (to KWS). We are grateful to Dr. Bruce L Patton and Dr. Yi-Ying Kuo for helpful discussions and comments on the manuscript.

## Additional information

### Competing interests

Ha H Truong, Niel M Henriksen: affiliated with Atomwise Inc. Show-Ling Shyng: Reviewing editor, *eLife*. The other authors declare that no competing interests exist.

### Funding

| Funder | Grant reference number | Author |
| --- | --- | --- |
| National Institute of General Medical Sciences | R01GM145784 | Show-Ling Shyng |
| Egyptian government predoctoral scholarship | GM1109 | Assmaa Elsheikh |

| Funder | Grant reference number | Author |
|---|---|---|
| Polish high-performance computing infrastructure PLGrid | PLL/2023/04/016512 | Katarzyna Walczewska-Szewc |
| National Institutes of Health | U24GM129547 | Show-Ling Shyng |

The funders had no role in study design, data collection and interpretation, or the decision to submit the work for publication.

### Author contributions

Assmaa Elsheikh, Data curation, Formal analysis, Investigation, Visualization, Methodology, Writing - original draft, Writing – review and editing; Camden M Driggers, Data curation, Formal analysis, Investigation, Visualization, Writing – review and editing; Ha H Truong, Conceptualization, Data curation, Formal analysis, Investigation, Methodology, Writing – review and editing; Zhongying Yang, Formal analysis, Investigation; John Allen, Investigation; Niel M Henriksen, Conceptualization, Project administration; Katarzyna Walczewska-Szewc, Resources, Data curation, Formal analysis, Investigation, Visualization, Methodology, Writing - original draft, Writing – review and editing; Show-Ling Shyng, Conceptualization, Data curation, Formal analysis, Supervision, Funding acquisition, Investigation, Visualization, Writing - original draft, Project administration, Writing – review and editing

### Author ORCIDs

Camden M Driggers http://orcid.org/0000-0002-2105-7175
Show-Ling Shyng https://orcid.org/0000-0002-8230-8820

Reviewer #1 (Public review): https://doi.org/10.7554/eLife.103159.3.sa1
Reviewer #2 (Public review): https://doi.org/10.7554/eLife.103159.3.sa2
Author response https://doi.org/10.7554/eLife.103159.3.sa3

## Additional files

### Supplementary files

Supplementary file 1. List of predicted binders that were subjected to biochemical and functional testing.

Supplementary file 2. Cryo-electron microscopy (cryoEM) and model statistics of the Aekatperone-bound SUR1/Kir6.2 structure.

MDAR checklist

### Data availability

The cryo-EM map has been deposited in the Electron Microscopy Data Bank (EMDB), and the coordinates have been deposited in the PDB under the following accession numbers: EMD-46820 and PDB ID 9DFX. Code and files for the MD simulation study are accessible via the following link: https://doi.org/10.18150/PSA6NC (*Walczewska-Szewc, 2025*). All data generated or analyzed during this study are included in the manuscript and supporting files.

The following datasets were generated:

| Author(s) | Year | Dataset title | Dataset URL | Database and Identifier |
|---|---|---|---|---|
| Driggers CM, ElSheikh A, Shyng S-L | 2025 | Cryo-EM structure of a SUR1/Kir6.2 ATP-sensitive potassium channel in the presence of Aekatperone in the closed conformation | https://www.ebi.ac.uk/emdb/EMD-46820 | EMDB, EMD-46820 |

*Continued on next page*

*Continued*

| Author(s) | Year | Dataset title | Dataset URL | Database and Identifier |
|---|---|---|---|---|
| Driggers CM, ElSheikh A, Shyng S-L | 2025 | Cryo-EM structure of a SUR1/Kir6.2 ATP-sensitive potassium channel in the presence of Aekatperone in the closed conformation | https://www.rcsb.org/structure/9dfx | RCSB Protein Data Bank, 9DFX |
| Walczewska-Szewc K | 2025 | Molecular Dynamics simulations of Aekatperone - New KATP Channel Pharmacochaperone | https://doi.org/10.18150/PSA6NC | RepOD, 10.18150/PSA6NC |

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
