## [Editor Report · eLife Assessment]

This **important** study demonstrates that screening by artificial intelligence can identify relevant novel compounds for interacting with KATP channels. The experimental work is **compelling**. The broader significance of this work relates to the possibility that KATP channel mutations linked to congenital hyperinsulinism may be effectively rescued to the cell surface with a drug, which could normalize insulin secretion or enhance the effectiveness of existing KATP channel activators such as diazoxide.

---

## [Referee Report · Reviewer #1 (Public review)]

Summary:

Multiple compounds that inhibit ATP-sensitive potassium (KATP) channels also chaperone channels to the surface membrane. The authors used an artificial intelligence (AI)-based virtual screening (AtomNet) to identify novel compounds that exhibit chaperoning effects on trafficking-deficient disease-causing mutant channels. One compound, which they named Aekatperone, acts as a low affinity, reversible inhibitor and effective chaperone. A cryoEM structure of KATP bound to Aekatperone showed that the molecule binds at the canonical inhibitory site.

Strengths and weaknesses:

The details of the AI screening itself are inevitably opaque, but appear to differ from classical virtual screening in not involving any physical docking of test compounds into the target site. The authors mention criteria that were used to limit the number of compounds, so that those with high similarity to known binders and 'sequence identity' (does this mean structural identity) were excluded. The identified molecules contain sulfonylurea-like moieties. How different are they from other sulfonylureas?

The experimental work confirming that Aekatperone acts to traffic mutant KATP channels to the surface and acts as a low affinity, reversible, inhibitor is comprehensive and clear, with very convincing cell biological and patch-clamp data, as is the cryoEM structural analysis, for which the group are leading experts. In addition to the three positive chaperone-effective molecules, the authors identified a large number of compounds that are predicted binders but apparently have no chaperoning effect.

The authors suggest that the novel compound may be a promising therapeutic for treatment of congenital hyperinsulinism due to trafficking defective KATP mutations. Because they are low affinity, reversible, inhibitors. This is a very interesting concept, and perhaps a pulsed dosing regimen would allow trafficking without constant channel inhibition (which otherwise defeats the therapeutic purpose), although it is unclear whether the new compound will offer advantages over earlier low-affinity sulfonylurea inhibitor chaperones. These include tolbutamide which has very similar affinity and effect to Aekatperone. As the authors point out this (as well as other sulfonlyureas) are currently out of favor because of potential adverse cardiovascular effects, but again, it is unclear why Aekatperone should not have the same concerns.

Comments on revised version:

The authors have been very responsive to the first reviews. No further comments.

---

## [Referee Report · Reviewer #2 (Public review)]

Summary:

In their study 'AI-based Discovery and CryoEM Structural Elucidation of a KATP Channel Pharmacochaperone', ElSheikh and colleagues undertake a computational screening approach to identify candidate drugs that may bind to an identified binding pocket in the SUR1 subunit of KATP channels. Other KATP channel inhibitors such as glibenclamide have been previously shown to bind in this pocket, and in addition to inhibition KATP channel function, these inhibitors can very effectively rescue cell surface expression of trafficking deficient KATP mutations that cause excessive insulin secretion (Congenital Hyperinsulinism). However, a challenge for their utility for treatment of hyperinsulinism has been that they are powerful inhibitors of the channels that are rescued to the channel surface. In contrast, successful therapeutic pharmacochaperones (eg. CFTR chaperones) permit function of the channels rescued to the cell membrane. Thus, a key criteria for the authors' approach in this case was to identify relatively low affinity compounds that target the glibenclamide binding site (and be washed off) - these could potentially rescue KATP surface expression, but also permit KATP function.

Strengths:

The main findings of the manuscript include:

(1) Computational screening of a large virtual compound library, followed by functional screening of cell surface expression, which identified several potential candidate pharmacochaperones that target the glibenclamide binding site.

(2) Prioritization and functional characterization of Aekatperone as a low affinity KATP inhibitor which can be readily 'washed off' in patch clamp, and cell based efflux assays. Thus the drug clearly rescues cell surface expression, but can be manipulated experimentally to permit function of rescued channels.

(3) Determination of the binding site and dynamics of this candidate drug by cryo-EM, and functional validation of several residues involved in drug sensitivity using mutagenesis and patch clamp.

The experiments are well-conceived and executed, and the study is clearly described. The results of the experiments are very straightforward and clearly support the conclusions drawn by the authors. I found the study to provide important new information about KATP chaperone effects of certain drugs, with interesting considerations in terms of ion channel biology and human disease.

Context and remaining challenges:

(1) The chaperones can effectively rescue KATP trafficking mutants, but clearly not as strongly as the higher affinity inhibitor glibenclamide. There is likely a challenging relationship between efficacy of trafficking rescue and channel inhibition (ie. rescued channels are inhibited and therefore non-functional) that will need to be overcome in terms of applying drugs of this class. This is recognized and clarified appropriately by the authors both in their experimental approaches and discussion. In experiments it is straightforward to wash off the chaperone, but this would not be the case in an organism.

(2) Recent developments with ion channel trafficking correctors in the CFTR field illustrate the importance of investigating underlying mechanisms. Development of pharmacological tools and approaches in other channel types (such as KATP or other transporters and channels) will build our understanding of pathways involved in regulating maturation of membrane proteins, and ways to manipulate them.

Comments on revised version:

I have no further suggestions, thank you for the detailed response.

---

## [Author Response]

The following is the authors’ response to the original reviews.

**Reviewer #1 (Public review):**
Summary:Multiple compounds that inhibit ATP-sensitive potassium (KATP) channels also chaperone channels to the surface membrane. The authors used an artificial intelligence (AI)-based virtual screening (AtomNet) to identify novel compounds that exhibit chaperoning effects on trafficking-deficient disease-causing mutant channels. One compound, which they named Aekatperone, acts as a low affinity, reversible inhibitor and effective chaperone. A cryoEM structure of KATP bound to Aekatperone showed that the molecule binds at the canonical inhibitory site.Strengths and weaknesses:The details of the AI screening itself are inevitably opaque, but appear to differ from classical virtual screening in not involving any physical docking of test compounds into the target site. The authors mention criteria that were used to limit the number of compounds, so that those with high similarity to known binders and 'sequence identity' (does this mean structural identity) were excluded. The identified molecules contain sulfonylurea-like moieties. How different are they from other sulfonylure4as?

We thank the reviewers for the questions. As part of the library preparation, molecules with greater than 0.5 Tanimoto similarity in ECFP4 space to any known binders of the target protein and its homologs within 70% sequence identity were excluded to increase the possibility of identifying novel hits. After scoring and ranking the molecules by the AtomNet technology, a diversity clustering was performed using the Butina algorithm (Butina D. Unsupervised Data Base Clustering Based on Daylight’s Fingerprint and Tanimoto Similarity: A Fast and Automated Way To Cluster Small and Large Data Sets, J. Chem. Inf. Comput. Sci. 1999, 39, 747–750) with a Tanimoto similarity cutoff of 0.35 in ECFP4 space to minimize selection of structurally similar scaffolds for the final compound buy-list. We have revised the results and methods sections to make this clear.

Sulfonylureas are defined by their core structure comprising a sulfonyl group (–S(=O)_2_) and a urea moiety (–NH–CO–NH–). While some compounds identified in our study contain a sulfonamide group (R-S(=O) _2_-NR_2_), they differ structurally from sulfonylureas by lacking the key urea group and by incorporating unique R-group substitutions (we have now added this to Figure 1A legend). For example, compound C27 (Z2068224500) includes a sulfonamide group but not a urea moiety. Likewise, C45 (Aekatperone, Z1620764636) contains a sulfonamide group along with an aromatic, nitrogen-rich heterocyclic ring, but no urea group. Additionally, the R-groups in these compounds are more complex than the simple aromatic or alkyl chains typical of sulfonylureas. They include heterocyclic aromatic systems and nitrogen-rich structures, which likely influence their binding properties and lipophilicity. These structural differences suggest distinct functional and pharmacological profiles as supported by our biochemical and functional studies.

The experimental work confirming that Aekatperone acts to traffic mutant KATP channels to the surface and acts as a low affinity, reversible, inhibitor is comprehensive and clear, with very convincing cell biological and patch-clamp data, as is the cryoEM structural analysis, for which the group are leading experts. In addition to the three positive chaperone-effective molecules, the authors identified a large number of compounds that are predicted binders but apparently have no chaperoning effect. Did any of them have inhibitory action on channels? If so, does this give clues to separating chaperoning from inhibitory effects?

This is an interesting question. Evidence from cryo-EM, biochemical and electrophysiology studies reveal a critical role of Kir6.2 N-terminus in K_ATP_ channel assembly and gating, and that pharmacological chaperones like glibenclamide, repaglinide, carbamazepine, and now aekatperone exert their chaperoning and inhibitory effects by stabilizing the interaction between Kir6.2 N-terminus and the SUR1-ABC core. This stabilization, while promoting the assembly of Kir6.2 and SUR1 to “chaperone” trafficking-impaired mutant channels to the cell surface, also inhibits the channel by restricting the Kir6.2 C-terminal domain from rotating to an open state. An additional mechanism by which these compounds inhibit channel activity is by preventing SUR1-NBD dimerization, which mediates physiological activation of the channel by MgADP (see review: Driggers CM, Shyng SL. Mechanistic insights on K_ATP_ channel regulation from cryo-EM structures. J Gen Physiol. 2023 Jan 2;155(1): e202113046, PMID: 36441147). From our compound screening, we did find some compounds that showed mild inhibition of the channel by electrophysiology but no obvious chaperone effects by western blots. It is possible that small chaperoning effects of some compounds showing mild channel inhibition effects were missed due to the lower sensitivity of the western blot assay compared to electrophysiology. Alternatively, these compounds could inhibit channels by preventing SUR1NBD dimerization without stabilizing the Kir6.2 N-terminus, which is required for the chaperone effect based on our model. Unfortunately, we did not find any compounds that show chaperone effects but no channel inhibition effects, which is consistent with our understanding of how this type of K_ATP_ chaperones work (i.e. by stabilizing Kir6.2 N-terminus interaction with SUR1’s ABC core).

The authors suggest that the novel compound may be a promising therapeutic for treatment of congenital hyperinsulinism due to trafficking defective KATP mutations. Because they are low affinity, reversible, inhibitors. This is a very interesting concept, and perhaps a pulsed dosing regimen would allow trafficking without constant channel inhibition (which otherwise defeats the therapeutic purpose), although it is unclear whether the new compound will offer advantages over earlier low-affinity sulfonylurea inhibitor chaperones. These include tolbutamide which has very similar affinity and effect to Aekatperone. As the authors point out this (as well as other sulfonlyureas) are currently out of favor because of potential adverse cardiovascular effects, but again, it is unclear why Aekatperone should not have the same concerns.

We thank the reviewer for the comments. This is clearly an important question to address in the future. While we have not directly tested the effects of Aekatperone on cardiac functions, we did assess its inhibitory effect on cells expressing the cardiac K_ATP_ channel isoform (SUR2A/Kir6.2). Our results indicate that Aekatperone exhibits higher sensitivity toward the pancreatic K_ATP_ channel isoform (SUR1/Kir6.2) compared to the cardiac isoform. However, we acknowledge that Aekatperone could still have cardiotoxic effects through its potential action on other channels, such as the hERG channel.

It is worth noting that tolbutamide, despite its known cardiotoxic effects, does not exert these effects through cardiac K_ATP_ channel inhibition. This has been demonstrated in studies showing no inhibitory effect of tolbutamide on SUR2A/Kir6.2 channels and on channels formed by Kir6.2 and SUR1 harboring the S1238Y mutation (also shown as S1237Y in some studies using a different SUR1 isoform)--the amino acid substitution found in SUR2A at the corresponding position (Ashfield R, Gribble FM, Ashcroft SJ, Ashcroft FM. Identification of the high-affinity tolbutamide site on the SUR1 subunit of the K_ATP_ channel. Diabetes. 1999 Jun;48(6):1341-7, PMID: 10342826). This suggests that tolbutamide’s cardiotoxic effects might involve other targets like the hERG channel. Interestingly, tolbutamide contains a hydrophobic tail and aromatic rings that align well with the structural features for hERG interaction (Garrido A, Lepailleur A, Mignani SM, Dallemagne P, Rochais C. hERG toxicity assessment: Useful guidelines for drug design. Eur J Med Chem. 2020 Jun 1;195:112290, PMID: 32283295). In contrast, highaffinity sulfonylureas such as glibenclamide and glimepiride, which have additional benzamide moieties, are associated with lower cardiovascular risks (Douros A, Yin H, Yu OHY, Filion KB, Azoulay L, Suissa S. Pharmacologic Differences of Sulfonylureas and the Risk of Adverse Cardiovascular and Hypoglycemic Events. Diabetes Care. 2017, 40:1506-1513, PMID: 28864502). Given these considerations, a comprehensive assessment of Aekatperone’s potential cardiotoxicity is crucial. Future studies involving in silico modeling, in vitro, and in vivo experiments will be essential to evaluate Aekatperone’s interaction with hERG and other offtarget effects. These efforts will help clarify its safety profile. This point has now been added to the Discussion.

**Reviewer #2 (Public review):**
Summary:In their study 'AI-based Discovery and CryoEM Structural Elucidation of a KATP Channel Pharmacochaperone', ElSheikh and colleagues undertake a computational screening approach to identify candidate drugs that may bind to an identified binding pocket in the SUR1 subunit ofKATP channels. Other KATP channel inhibitors such as glibenclamide have been previously shown to bind in this pocket, and in addition to inhibition KATP channel function, these inhibitors can very effectively rescue cell surface expression of trafficking deficient KATP mutations that cause excessive insulin secretion (Congenital Hyperinsulinism). However, a challenge for their utility for treatment of hyperinsulinism has been that they are powerful inhibitors of the channels that are rescued to the channel surface. In contrast, successful therapeutic pharmacochaperones (eg. CFTR chaperones) permit function of the channels rescued to the cell membrane. Thus, a key criteria for the authors' approach in this case was to identify relatively low affinity compounds that target the glibenclamide binding site (and be washed off) - these could potentially rescue KATP surface expression, but also permit KATP function.Strengths:The main findings of the manuscript include:(1) Computational screening of a large virtual compound library, followed by functional screening of cell surface expression, which identified several potential candidate pharmacochaperones that target the glibenclamide binding site.(2) Prioritization and functional characterization of Aekatperone as a low affinity KATP inhibitor which can be readily 'washed off' in patch clamp, and cell based efflux assays. Thus the drug clearly rescues cell surface expression, but can be manipulated experimentally to permit function of rescued channels.(3) Determination of the binding site and dynamics of this candidate drug by cryo-EM, and functional validation of several residues involved in drug sensitivity using mutagenesis and patch clamp.The experiments are well-conceived and executed, and the study is clearly described. The results of the experiments are very straightforward and clearly support the conclusions drawn by the authors. I found the study to provide important new information about KATP chaperone effects of certain drugs, with interesting considerations in terms of ion channel biology and human disease.Weaknesses:I don't have any major criticisms of the study as described, but I had some remaining questions that could be addressed in a revision.(1) The chaperones can effectively rescue KATP trafficking mutants, but clearly not as strongly as the higher affinity inhibitor glibenclamide. Is this relationship between inhibitory potency, and efficacy of trafficking an intrinsic challenge of the approach? I suspect that it may be an intractable problem in the sense that the inhibitor bound conformation that underlies the chaperone effect cannot be uncoupled from the inhibited gating state. But this might not be true (many partial agonist drugs with low efficacy can be strongly potent, for example). In this case, the approach is really to find a 'happy medium' of a drug that is a weak enough inhibitor to be washed away, but still strong enough to exert some satisfactory chaperone effect. Could some additional clarity be added in the discussion on whether the chaperone and gating effects can be 'uncoupled'.

Thank you for the suggestion. A similar question was raised by Reviewer 1, which was addressed above (public review, point 2). We have now added more discussion to clarify this point.

(2) Based on the western blots in Figure 2B, the rescue of cell surface expression appears to require a higher concentration of AKP compared to the concentration response of channel inhibition (~9 microM in Figure 3, perhaps even more potent in patch clamp in Figure 2C). Could the authors clarify/quantify the concentration response for trafficking rescue?

Thank you for bringing up this observation. Indeed, the pharmacochaperone effects of Aekatperone as well as other previously published K_ATP_ pharmacochaperones require higher concentrations compared to their inhibitory effects on surface-expressed channels. This difference likely stems from the necessity for these compounds to cross the cell membrane and interact with newly synthesized channels in the endoplasmic reticulum, where the trafficking rescue occurs. We estimate that effective pharmacochaperone activity for Aekatperone can be achieved at concentrations ranging from 50 to 100 µM in cells expressing trafficking-deficient K_ATP_ channel mutants, higher than that required for inhibition of surface-expressed channels (~9 µM IC50). Future work could focus on medicinal chemistry modifications, for example esterification of Aekatperone (Zhou G. Exploring Ester Prodrugs: A Comprehensive Review of Approaches, Applications, and Methods. Pharmacology & Pharmacy, 2024, 15, 269-284). Once inside the cell, the esters would be cleaved by endogenous esterases to release the active compound, ensuring efficient intracellular delivery. This strategy could potentially improve membrane permeability and bioavailability of the compound, which would lower the required concentrations to achieve desired chaperoning effects.

(3) A future challenge in the application of pharmacochaperones of this type in hyperinsulinism may be the manipulation of chaperone concentration in order to permit function. In experiments it is straightforward to wash off the chaperone, but this would not be the case in an organism. I wondered if the authors had attempted to rescue channel function with diazoxide ine presence of AKP, rather than after washing off (ie. is AKP inhibition insurmountable, or can it be overcome by sufficient diazoxide).

Thank you for raising this important point. We have previously shown (Martin GM et al. Pharmacological Correction of Trafficking Defects in ATP-sensitive Potassium Channels Caused by Sulfonylurea Receptor 1 Mutations. J Biol Chem. 2016, 291: 21971-21983, PMID: 27573238) that diazoxide, which stabilizes K_ATP_ channels in an open conformation, also reduces physical association between Kir6.2 N-terminus and SUR1 as demonstrated by reduced crosslinking of engineered azido-phenylalanine (an unnatural amino acid) at Kir6.2 N-terminal amino acid 12 position to SUR1. Incubating cells with diazoxide did not rescue the trafficking mutants but actually further reduced the maturation efficiency of trafficking mutants. For this reason, we did not include diazoxide during Aekatperone incubation and instead added diazoxide after Aekatperone washout to potentiate the activity of mutant channels rescued to the cell surface. In vivo, we envision testing alternating Aekatperone and diazoxide dosing to maximize functional rescue of K_ATP_ trafficking mutants.

(4) Do the authors have any information about the turnover time of KATP after washoff of the chaperone (how stable are the rescued channels at the cell surface)? This is a difficult question to probe when glibenclamide is used as a chaperone, but maybe much simpler to address with a lower affinity chaperone like AKP.

Thank you for your thoughtful comment. While we have not yet tested the duration of rescued K_ATP_ channels at the cell surface following Aekatperone washout, we have conducted similar studies with carbamazepine (Chen PC et al. Carbamazepine as a novel small molecule corrector of trafficking-impaired ATP-sensitive potassium channels identified in congenital hyperinsulinism. J Biol Chem. 2013, 288: 20942-20954, PMID: 23744072), another compound exhibiting reversible inhibitory and chaperone effects (apparent affinity between glibenclamide and Aekatperone). Our previous findings with carbamazepine showed that in cultured cells its chaperone effects were detectable as early as 1 hour and peaked around 6 hours after treatment. Furthermore, when carbamazepine was removed following a 16-hour treatment, the rescue effect persisted for up to 6 hours post-drug removal. These results provide a potential duration of the surface expression rescue effects of reversible pharmacochaperones.

**Reviewer #1 (Recommendations for the authors):**
The paper is well-written and comprehensive with only very minor essentially copy-editing needed. That said, it would be good if the authors could answer the main points raised above:(1) What is the relevant Tanimoto parameters and sequence identity (does this mean structural identity) for the identified compounds?

As we answered above in response to the overall assessment, to facilitate the identification of novel hits, molecules with greater than 0.5 Tanimoto similarity in ECFP4 space to any known binders of the target protein and its homologs within 70% amino acid sequence identity were excluded from the commercial library. Additionally, after scoring and ranking the molecules by the AtomNet technology, a diversity clustering was performed on the top 30,000 molecules using the Butina algorithm with a Tanimoto similarity cutoff of 0.35 in ECFP4 space to minimize selection of structurally similar scaffolds for the final compound buy-list.

(2) Did any of the identified putative binders have inhibitory action on channels? If so, does this give clues to separating chaperoning from inhibitory effects?

Please see response to the same question in the overall assessment above.

(3) Acknowledge that the identified compounds contain sulfonylurea-like moieties, and address why Aekatperone should (or perhaps does not) offer anything advantage over low affinity sulfonrylureas such as tolbutamide?

Please see response to the same question in the overall assessment above.

**Reviewer #2 (Recommendations for the authors):**
Thank you for assembling the interesting study, which I felt was well designed and communicated. The diverse approaches used in the study, with consistent findings, were definitely a strength. The core findings are also well distilled in the main body of the text, and although there is quite a lot of supplementary information, I felt that it was presented appropriately and well selected in terms of what would be important for readers hoping to learn more. In addition to the questions described above, I only had a few minor editorial issues that could be fixed related to presentation.(1) Figure 1B. The colours and resolution of the chemical structures are difficult to see clearly and could be improved.

We have revised the figure accordingly.

(2) This is a minor wording point... first sentence of the discussion describes the drugs as pancreatic-selective, when it would be more clear to describe them as selective for the pancreatic isoform of KATP (Kir6.2/SUR1), or perhaps better as 'exhibiting ~4-5 fold selective for SUR1-containing KATP channels vs. SUR2A or SUR2B'.

We have changed the wording as suggested.

(3) As a curiosity (not necessary to do more experiments), but I am curious if the authors know whether there is any meaningful enhancement of trafficking of WT channels by AKP.

All pharmacochaperones we have identified to date including Aekatperone also slightly enhance WT channel surface expression (10-20%).

**Reviewing editor recommendations:**
(1) Given the modest resolution of the EM reconstruction, it is perhaps not entirely clear how AKP was assigned to the density observed. Specifically, it would be helpful to include a comparison of an AKP-free map and the current AKP map (filtered to a similar resolution) showing slice views of densities in the region around the inferred binding site. This would be very helpful in ascertaining whether the cryoEM reconstruction is an independent validation of the computational and functional experiments or whether the density inference depends on the additional knowledge.

We appreciate the editor’s suggestion. We have now added a Supplemental Figure (Supplementary Figure 7 in the revised manuscript) that compares our AKP-free cryoEM density deposited previously to the EMDB (EMD-26320) and the AKP-bound cryoEM density from this study, with cryoEM density (filtered to the same resolution) superimposed on the structural model.

(2) It could help to mention in brief what is a probable mechanism of AKP inhibition - that is how after binding of AKP, channel opening is restricted. Is it similar to that of other site A ligands?

Based on the strong Kir6.2 N-terminal cryoEM density observed in our AKP map, AKP most likely inhibits K_ATP_ channels by trapping the Kir6.2 N-terminus in the central cavity of SUR1’s ABC core thus preventing Kir6.2-C-terminal domain from rotating to an open conformation, similar to other ligands that stabilize the Kir6.2 N-terminus-SUR1 interface by binding to site A (such as tolbutamide and AKP), site B (such as repaglinide), or both site A and site B (such as glibenclamide). We have now included this in the revised Results and Discussion sections.

(3) In the context of the MD simulations, do other site A ligands (which from my understanding bind at a similar site) also exhibit similar flexibility as AKP? If there is information available on the flexibility of ligands of varying affinities, bound to the same site, maybe some correlative inferences can be drawn? However, in MD simulation trajectories it is not entirely uncommon for a ligand to simply get trapped in a local energy well. Since the authors have performed significant analysis of their MD results it could be worth mentioning/discussing such phenomena.

Previously published MD data addressing ligand dynamics, such as glibenclamide in the SUR1 pocket (Walczewska-Szewc K, Nowak W. Photo-Switchable Sulfonylureas Binding to ATPSensitive Potassium Channel Reveal the Mechanism of Light-Controlled Insulin Release. J Phys Chem B. 2021, 125: 13111-13121, PMID: 34825567), indicate a certain degree of flexibility. Unfortunately, we cannot directly compare these results, as the simulations were performed without the KNtp domain in the SUR1 cavity, which partially contributes to ligand stabilization. This is an issue we plan to investigate in the future.

In this study, we ran five independent MD simulations, each 500 ns long, resulting in a total of 2.5 μs of simulation time. Across all replicates, the ligand stayed in the same position, with variations mainly in the dynamics of the blurred segment. Considering the length of the simulations and the consistency across the runs, we believe this binding pose is stable and represents a global (or at least highly stable) energy minimum, consistent with the cryo-EM data.

(4) In electrophysiological assays, 10 uM AKP seems to inhibit all currents (Figure 2), but in the Rb+ flux assay ~10 uM appears to be the IC50. The reason for this difference is not entirely clear and it would help to comment on this.

Thank you for noticing the difference. The initial electrophysiological experiments were conducted using the very small amount of AKP provided to us from Atomwise. We estimated the concentration of the reconstituted AKP the best we could, but the concentration was likely to not be very accurate due to difficulty in handling the very small amount of the AKP powder. Subsequent Rb^+>/sup> efflux experiments were conducted using a different, larger batch of AKP we purchased from Enamine. We have now stated this in the Methods section.^